# Differential expression of tissue-restricted antigens among mTEC is associated with distinct autoreactive T cell fates

Marie-Ève Lebel [1,2], Marie Coutelier [3,4], Maria Galipeau [1], Claudia L. Kleinman [3,4], James J. Moon [5] & Heather J. Melichar [1,6] ✉

Medullary thymic epithelial cells (mTEC) contribute to the development of T cell tolerance by expressing and presenting tissue-restricted antigens (TRA), so that developing T cells can assess the self-reactivity of their antigen receptors prior to leaving the thymus. mTEC are a heterogeneous population of cells that differentially express TRA. Whether mTEC subsets induce distinct autoreactive T cell fates remains unclear. Here, we establish bacterial artificial chromosome (BAC)-transgenic mouse lines with biased mTEC$^{lo}$ or mTEC$^{hi}$ expression of model antigens. The transgenic lines support negative selection of antigen-specific thymocytes depending on antigen dose. However, model antigen expression predominantly by mTEC$^{lo}$ supports TCRαβ$^+$ CD8αα intraepithelial lymphocyte development; meanwhile, mTEC$^{hi}$-restricted expression preferentially induces T$_{reg}$ differentiation of antigen-specific cells in these models to impact control of infectious agents and tumor growth. In summary, our data suggest that mTEC subsets may have a function in directing distinct mechanisms of T cell tolerance.

[1] Maisonneuve-Rosemont Hospital Research Center, 5415 Boulevard de l'Assomption, Montreal, QC H1T 2M4, Canada. [2] Département de microbiologie, infectiologie et immunologie, Université de Montréal, C.P. 6128, succ. Centre-ville, Montreal, QC H3C 3J7, Canada. [3] The Lady Davis Institute for Medical Research, Jewish General Hospital, 3999 Côte Ste-Catherine Road Room E-542, Montreal, QC H3T 1E2, Canada. [4] Department of Human Genetics, McGill University, Rm W-315, Strathcona Anatomy & Dentistry Building 3640 rue University, Montreal, QC H3A 0C7, Canada. [5] Center for Immunology and Inflammatory Diseases and Division of Pulmonary and Critical Care Medicine, Massachusetts General Hospital and Harvard Medical School, 149 13th Street, Charlestown, MA 02129, USA. [6] Département de médecine, Université de Montréal, C.P. 6128, succ. Centre-ville, Montreal, QC H3C 3J7, Canada. ✉email: heather.melichar@umontreal.ca

T cell tolerance to self-antigen is established, in part, con-currently with T cell development in the thymus. Devel-oping T cells bearing antigen receptors with high reactivity to self-antigen encountered in the thymus can be deleted via negative selection, diverted towards immunoregulatory lineages, or rendered unresponsive[1–3]. A significant proportion of negative selection occurs in response to ubiquitously expressed self-antigen in the thymic cortex, while the medullary micro-environment plays an important role in the generation of T cell tolerance to tissue restricted antigens (TRA)[4–6]. The promiscuous nature of gene expression within the medullary thymic epithelial cell (mTEC) population provides an opportunity for developing T cells to test the self-reactivity of their antigen receptor against peptides derived from ~90% of all genes[7].

Various T cell-extrinsic parameters that affect the strength of T cell receptor (TCR) signals play an important role in tolerance induction in the thymus. For instance, the frequency of major histocompatibility complex (MHC) class II$^+$ cells expressing a given TRA in the thymus correlates with the mechanism of CD4$^+$ T cell tolerance; it has been suggested that high numbers of cells expressing self-antigen drive clonal deletion whereas lower levels of thymic antigen presentation predominantly leads to non-deletional tolerance mechanisms and regulatory T cell (T$_{reg}$) generation[8]. Modulation of co-stimulatory signals also affects tolerance induction. Co-stimulation is required for negative selection of MHC class II-restricted autoreactive thymocytes as well as T$_{reg}$ development[9,10]. Further, the presence or absence of co-stimulation discriminates between clonal deletion and TCRαβ$^+$ CD8α$^+$CD8β$^-$ (CD8αα) intraepithelial lymphocyte (IEL) lineage diversion in thymocytes bearing antigen receptors with high-affinity for self-peptides[11].

It is becoming increasingly clear that there is significant het-erogeneity among mTEC and that TRA are differentially expressed among these subsets[12–16]. mTEC subsets can be dis-tinguished by differences in MHC and co-stimulatory molecule expression. The mTEC$^{lo}$ (CD80/CD86$^{lo}$ MHC class II$^{lo}$) com-partment includes "immature" progenitor cells, as well as a functionally mature subset and a population of terminally dif-ferentiated epithelial cells[12,13,17]. As they differentiate, immature mTEC$^{lo}$ upregulate co-stimulatory molecule expression and MHC class II levels to become mTEC$^{hi}$[18,19]. mTEC$^{hi}$ include a subset that expresses autoimmune regulator (Aire), a well-known driver of TRA expression that also plays pleiotropic roles in T cell tolerance[20]. While Aire-driven TRA expression is restricted to the mature mTEC$^{hi}$ compartment, Aire-independent TRA are also expressed by the mTEC$^{lo}$ subset[21]. Therefore, different mTEC populations likely support tolerance to distinct TRA pools, but whether these subsets play unique roles in the quality or mechanism of T cell tolerance is not yet clear.

To decipher whether self-antigen expression by different mTEC subsets contributes to distinct mechanisms of T cell tol-erance, we describe new transgenic (Tg) mice in which Aire-dependent and Aire-independent TRA promoters drive distinct patterns of model antigen expression among mTEC. We observe that autoreactive T cell diversion into immunoregulatory T cell lineages is influenced by the mTEC subset expressing the model antigen in these models with functional consequences. This is particularly relevant to the design of therapeutic approaches that aim to modulate the function of self-reactive T cells.

## Results

### Distinct patterns of model antigen expression among mTEC.
To characterize T cell tolerance to TRA differentially expressed among mTEC subsets, we generated new Tg mouse models. In these mice, expression of a model antigen is directed from either the *C-reactive protein* (*Crp*) locus, whose Aire-independent expression is preferentially detected among immature mTEC$^{lo}$, or the *Insulin2* (*Ins2*) locus, where expression is restricted to Aire$^+$ mTEC$^{hi}$ (Supplementary Fig. 1a, b[12,21]). To maintain many of the physiological elements that regulate expression from these loci without interfering with gene function, we engineered ~200 kb bacterial artificial chromosomes containing the locus of interest to include a modified ovalbumin (OVA) gene at the start codon of the *Crp* and *Ins2* genes (Fig. 1a). The modified OVA gene contains additional MHC class I and II epitopes (LCMV gp33, LCMV gp66, and 2W), referred to as a "*Universal Self-Antigen*" (*USA*), is flanked by the Kb leader sequence and Db transmembrane region[22], and is followed by a sequence encoding for a self-cleaving peptide and GFP; the construct is designed such that OVA and the additional epitopes are in the extracellular domain.

We assessed the fidelity of *USA* expression in sorted mTEC populations from *Ins2*- and *Crp*-promoter driven Tg mice (INS2 and CRP Tg mice, respectively) (Fig. 1b, c and Supplementary Fig. 1a). In INS2 Tg mice, *USA* mRNA is restricted to mTEC$^{hi}$ and is dependent on Aire, whereas *USA* mRNA is more abundantly expressed in the mTEC$^{lo}$ subset in CRP Tg mice on both wild-type (WT) and *Aire*-deficient backgrounds (Fig. 1b). We screened *USA* expression in multiple Tg founder lines and maintained two independent lines of each for further analysis; the CRP$^{lo}$ and INS2$^{lo}$ Tg founder lines have lower levels of *USA* mRNA expression (Fig. 1c). Next, because *USA* expression was difficult to detect in total mTEC in some Tg lines, we estimated total expression based on qRT-PCR analysis of the mTEC$^{lo}$ and mTEC$^{hi}$ subsets. More specifically, the relative expression values in mTEC$^{lo}$ and mTEC$^{hi}$ were multiplied by the proportion of each population to obtain an estimation of the expression in total mTEC. There is limited variability in expression among INS2 Tg founder lines; INS2$^{lo}$ Tg express ~2-fold lower levels of *USA* mRNA as compared to INS2 Tg mice (Fig. 1d). Overall, the INS2$^{lo}$ and CRP$^{lo}$ Tg founder lines have similar levels of *USA* expression in the total mTEC compartment, although *USA* mRNA is preferentially expressed among mTEC$^{lo}$ in the CRP$^{lo}$ Tg mice, while it is restricted to mTEC$^{hi}$ in the INS2$^{lo}$ Tg mice. In addition, *USA* is not detected in B cells, macrophages nor dendritic cells isolated from the thymus of the INS2 or CRP Tg mice (Supplementary Fig. 1c, d). In the periphery, model antigen is expressed in the liver of the CRP Tg mice and in the islet-enriched fraction of the pancreas of INS2 Tg mice (Supplementary Fig. 1e).

Apart from the amount of co-stimulatory molecules expressed by model antigen positive cells, the number of TRA expressing cells as well as TRA expression level on a per cell basis could also impact the signal received by developing thymocytes and, thus, their fate. While it is well known that Aire-dependent TRAs such as INS2 are usually expressed by only a small proportion of mTEC$^{hi}$ cells, the expression pattern of Aire-independent TRAs genes such as *Crp* is less well characterized. To further identify differences in *Crp* and *Ins2* expression and characterize transgene expression, we performed single-cell RNA sequencing analysis on ~10,000 total sorted thymic epithelial cells (TEC) from CRP, CRP$^{lo}$, and INS2 Tg mice (Fig. 1e–i and Supplementary Fig. 1f–h). We detect *USA* expression in TEC from CRP Tg mice. However, transgene expression is not detected in TEC from CRP$^{lo}$ Tg mice and is very low in TEC isolated from INS2 Tg mice reflecting the differences we observe in the bulk mTEC population from these Tg mice. Although the number of cells expressing the transgene (*USA*) in CRP Tg mice is lower than the number of endogenous *Crp*-positive cells (3.2% vs 36.6%), their distribution across cell types is similar, with a marked predominance in mTEC$^{lo}$ cells (85.9% and 84.3%, respectively) (Fig. 1g). Endogenous *Crp* and

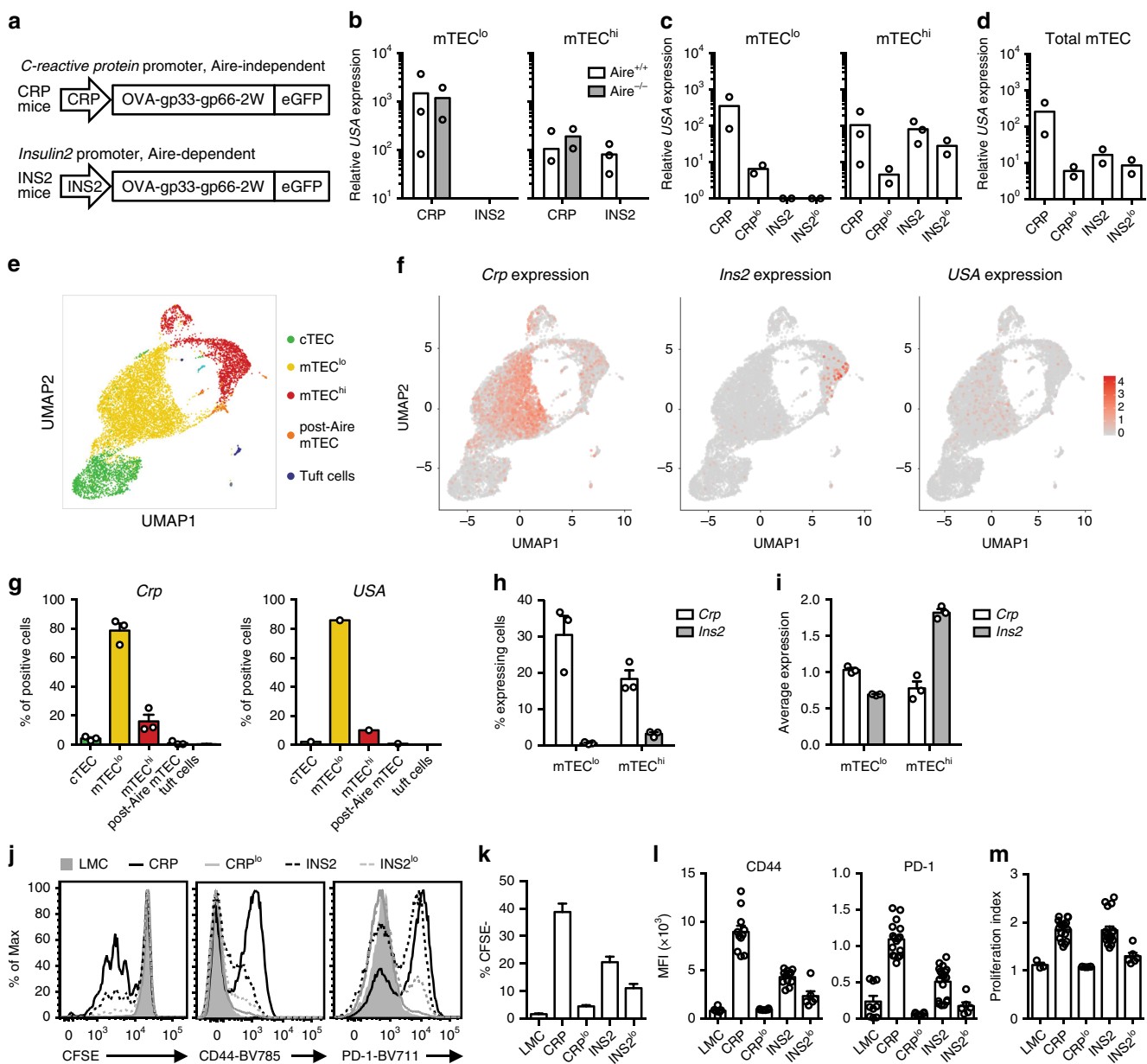

**Fig. 1 *Crp* and *Ins2* promoters drive model antigen expression in distinct patterns among mTEC subsets. a** Tg mouse constructs. **b** Relative *USA* expression in sorted mTEC$^{lo}$ and mTEC$^{hi}$ from *Aire$^{+/+}$* or *Aire$^{-/-}$* CRP and INS2 Tg mice. **c** *USA* mRNA expression in mTEC$^{lo}$ and mTEC$^{hi}$ from CRP, CRP$^{lo}$, INS2 and INS2$^{lo}$ Tg mice. **d** Calculation of *USA* mRNA expression in total mTEC. The thymus from three mice were pooled together prior to sorting, and data are representative of at least two independent experiments. **e–i** Single-cell RNA sequencing of total thymic epithelial cells (TEC) from CRP, CRP$^{lo}$, and INS2 Tg mice. **e** Uniform Manifold Approximation and Projection (UMAP) of TEC from CRP Tg mice. **f** UMAP highlighting *Crp*, *Ins2*, and *USA* transgene expression in TEC from CRP Tg mice. **g** Distribution of the *Crp*- or *USA*-expressing cells among the different TEC clusters isolated from the three Tg mice (*Crp*) or from CRP Tg mice (*USA*). **h** Proportion of mTEC$^{lo}$ and mTEC$^{hi}$ cells that express *Crp* or *Ins2* from the three Tg mice. **i** Average log-normalized per cell expression of *Crp* and *Ins2* among mTEC$^{lo}$ and mTEC$^{hi}$ isolated from the three Tg mice. Data from **g–i** are from one experiment with one mouse per Tg line and are represented as mean ± SEM for analysis of *Crp* and *Ins2*. **j** Representative histograms of mature OT-I T cell proliferation and activation 72 h after overlay on the indicated thymic slices as indicated by CFSE dilution as well as CD44 and PD-1 expression. **k** Compilation of the proportion of CFSE$^{lo}$ OT-I T cells 72 h after overlay on the indicated thymic slices (mean ± SEM, n = 12 LMC, 23 CRP, 15 CRP$^{lo}$, 24 INS2, 15 INS2$^{lo}$, from 3 independent experiments). **l** Compilation of the mean fluorescence intensity (MFI) of CD44 and PD-1 and **m** the proliferation index of the proliferating mature OT-I T cells. Littermate control (LMC) (mean ± SEM, n = 3 LMC, 17 CRP, 6 CRP$^{lo}$, 18 INS2, 6 INS2$^{lo}$, from 3 independent experiments). Source data are provided as a Source Data file.

*Ins2* expression at a single-cell level in TEC isolated from the three Tg mice reveals differences between these two TRA in the mTEC$^{hi}$ subset; *Crp* is expressed in a greater proportion of mTEC$^{hi}$, but at lower levels on a per cell basis while endogenous *Ins2* is expressed in very few mTEC$^{hi}$ cells, but at a relatively higher level in individual cells (Fig. 1h, i). In addition, the higher

endogenous *Crp* and CRP transgene expression in bulk mTEC$^{lo}$ vs mTEC$^{hi}$ as determined by qPCR (Supplementary Fig. 1b and Fig. 1b, c), is further explained by the single-cell RNA sequencing data. A higher proportion of mTEC$^{lo}$ express *Crp*, but *Crp* expression in individual cells is similar between these two mTEC subtypes (Fig. 1h, i); the same is true for *USA* expression among

mTEC$^{lo}$ and mTEC$^{hi}$ in the CRP Tg mouse (3.2% of mTEC$^{lo}$ and 1.2% of mTEC$^{hi}$ express the USA transgene with an average expression of 0.7 and 0.6, respectively). These analyses demonstrate differences in the expression patterns of two TRAs in TECs and suggest that, although CRP and INS2 transgene expression is detected in fewer cells, the broad pattern of expression is similar to that of their endogenous genes counterparts.

As a read-out of the quantity and quality of thymic model antigen presentation in the CRP and INS2 Tg mice, mature, model-antigen specific OT-I TCR Tg T cells isolated from lymph nodes were overlaid on thymic slices from the TRA Tg mouse lines, and their proliferation and activation state were measured 72 h later. We observe differences in the proportion of activated, proliferating OT-I TCR Tg T cells dependent on USA expression levels in the thymus (Fig. 1j–m and Supplementary Fig. 1i). INS2-driven USA mRNA expression in mTEC$^{hi}$ leads to a reduced proportion of OT-I TCR Tg cells that proliferate and a lower level of CD44 and PD-1 expression as compared to the CRP-driven USA expression (Fig. 1j–l). However, the cells that proliferate do so to a similar extent as indicated by the proliferation index (Fig. 1m). Despite similar overall levels of USA mRNA expression in the CRP$^{lo}$ and INS2$^{lo}$ Tg strains (Fig. 1d), CRP$^{lo}$ Tg thymic slices fail to induce significant proliferation or activation of the antigen-specific CD8$^+$ T cells which may be due to the low levels of co-stimulatory molecules in the mTEC$^{lo}$ population and the low level of model antigen expression in the mTEC$^{hi}$ compartment. Altogether, these data show that the four TRA Tg mouse models exhibit biased mTEC$^{lo}$ or mTEC$^{hi}$ expression of a model antigen.

**Negative selection of CD8 lineage T cells in Tg mice**. To analyze the extent of deletion of model antigen-specific MHC class I-restricted cells, we first tracked the development of OVA-specific OT-I TCR Tg Rag-deficient (OT-I) cells in USA expressing Tg mice. To limit competition for cognate antigen in these models, we used two different approaches. We first assessed negative selection using organotypic culture of thymic slices from CRP and INS2 Tg mice overlaid with a 1:1 mix of total CFSE-labeled OT-I and B6.SJL (CD45.1) control thymocytes. Despite significantly higher expression of the model antigen in the CRP Tg mice, the extent of deletion of high-affinity thymocytes after 24 h is similar in both CRP and INS2 Tg thymic slices as evidenced by a reduction in the proportion of OT-I:control thymocytes as compared to littermate controls (LMC) (Fig. 2a, b and Supplementary Fig. 2a) and the lower percentage of CD8$^+$ single positive (SP) thymocytes within the remaining OT-I TCRβ$^{hi}$ population (Fig. 2c). Deletion of OT-I thymocytes is inefficient on thymic slices from CRP$^{lo}$ and INS2$^{lo}$ Tg mice. In line with its Aire-dependency (Fig. 1b), the deletion of the OT-I TCR Tg thymocytes is completely abrogated in Aire$^{-/-}$ INS2 thymic slices, whereas deletion of antigen-specific thymocytes on CRP Tg thymic slices is unaffected by the absence of Aire (Fig. 2d). To confirm these results in vivo, we generated low-density bone marrow chimeras (BMC) in which LMC, CRP and INS2 Tg recipient mice were reconstituted with 1% OVA-specific OT-I or gp33-specific P14 TCR Tg TCRα knock-out (P14) and 99% WT bone marrow. We observe significant deletion of high-affinity OVA-specific OT-I cells in both the thymus and lymph nodes of CRP and INS2 Tg BMC as compared to LMC (Fig. 2e, f and Supplementary Fig. 2b, c). In addition, we detect partial deletion of P14 Tg thymocytes in the low-density BMC possibly reflecting either the lower affinity of the P14 TCR for its cognate peptide and/or less efficient processing and presentation of gp33 in this context (Supplementary Fig. 2d). These results are also consistent in CRP and INS2 Tg mice crossed to OT-I TCR Tg mice

(Supplementary Fig. 2e). Together, these results suggest that negative selection of MHC class I-restricted thymocytes depends on thymic antigen level in these models.

**mTEC$^{lo}$ support TCRαβ$^+$ CD8αα IEL differentiation**. Given the important role of co-stimulation in directing clonal deletion versus diversion toward the TCRαβ$^+$ CD8αα IEL pathway, we assessed the phenotype of the surviving OT-I thymocytes when cognate antigen is preferentially expressed in the mTEC$^{lo}$ (CD80/86$^{lo}$) versus mTEC$^{hi}$ (CD80/86$^{hi}$) compartments. We find a higher proportion of TCRβ$^{hi}$ OT-I DN cells in CRP but not INS2 Tg thymic slices (Fig. 3a, b and Supplementary Fig. 2a), and a large portion of TCRβ$^{hi}$ OT-I DN thymocytes express both PD-1 and CD122 in the CRP Tg thymic slices (Fig. 3c, d), markers of IEL precursors (IELp). In addition, a greater proportion of the TCRβ$^{hi}$ OT-I thymocytes are DN in the low-density CRP Tg BMC as compared to INS2 Tg and LMC chimeric mice (Fig. 3e, f and Supplementary Fig. 2b). In fact, a higher absolute number of CD8αα IELp is observed in the CRP Tg mice despite the significant depletion of OT-I cells in this model (Fig. 3f). While the proportion of OT-I T cells is greatly reduced in the thymus and peripheral lymph nodes from CRP and INS2 Tg BMC (Fig. 2e, f), a significant accumulation of these cells is observed in the IEL from the small intestine of the CRP Tg mice, and they are almost exclusively TCRαβ$^+$ CD8αα IELs (Fig. 3g–i). To rule out the possibility that generation of this population was due to the significantly higher level of USA mRNA expression in the CRP Tg mice as compared to the INS2 Tg mice, we repeated these experiments in the CRP$^{lo}$ Tg founder line. Interestingly, the lower USA transgene expression also induces an accumulation of TCRαβ$^+$ CD8αα OT-I IEL as compared to LMC (Fig. 3j, k) though OT-I thymocytes are not deleted (Fig. 3l). Notably, an increase in TCRβ$^{hi}$ DN thymocytes and TCRαβ$^+$ CD8αα IEL is also observed in low-density P14 Tg BMC in CRP, but not LMC or INS2 Tg mice (Supplementary Fig. 2f–j). In addition, we tested whether USA expression by radiation resistant cells is sufficient to generate TCRαβ$^+$ CD8αα IELs in this model by generating BMC where CRP Tg mice were irradiated and reconstituted with 99% β2 M$^{-/-}$ and 1% OT-I β2 M$^{-/-}$ bone marrow. Indeed, in CRP Tg mice in which MHC class I expression is restricted to radiation resistant cell populations, OT-I T cells differentiate into TCRαβ$^+$ CD8αα IELs and accumulate in the small intestine (Supplementary Fig. 2k, l). Furthermore, no increase in TCRαβ$^+$ CD8αα IEL differentiation is observed 6 weeks after an adoptive transfer of mature OT-I T cells into CRP Tg mice as compared to LMC suggesting a thymic origin of these cells (Supplementary Fig. 2m, n). Again, similar results are obtained regarding preferential CD8αα IEL development in CRP versus INS2 Tg mice when they are crossed onto the OT-I TCR Tg background (Supplementary Fig. 2o, p).

It has been suggested that the IELp transit through a CD4$^+$CD8$^+$ double positive (DP) intermediate in the thymus. However, it is not clear to what extent IELp selection occurs in the medulla. Both CCR7$^+$ post-selection DP thymocytes and more mature CD8$^+$ SP thymocytes could encounter self-antigen on mTEC$^{lo}$ cells. Using reporter mice in which developmentally distinct thymocyte subsets express YFP after CD4, distal Lck, or E8i promoter driven Cre mediated excision of a 5′ stop codon (CD4-YFP, distal Lck-YFP, and E8i-YFP, respectively), we sought to determine the developmental intermediates that give rise to thymic IELp. In CD4-YFP mice in which YFP expression is induced at the DP stage[23], almost all IELp are YFP$^+$, suggesting that they progress through a DP stage before differentiating into IELp (Supplementary Fig. 3a–c). In the distal Lck-YFP and the E8i-YFP mice, in which YFP expression is induced after positive

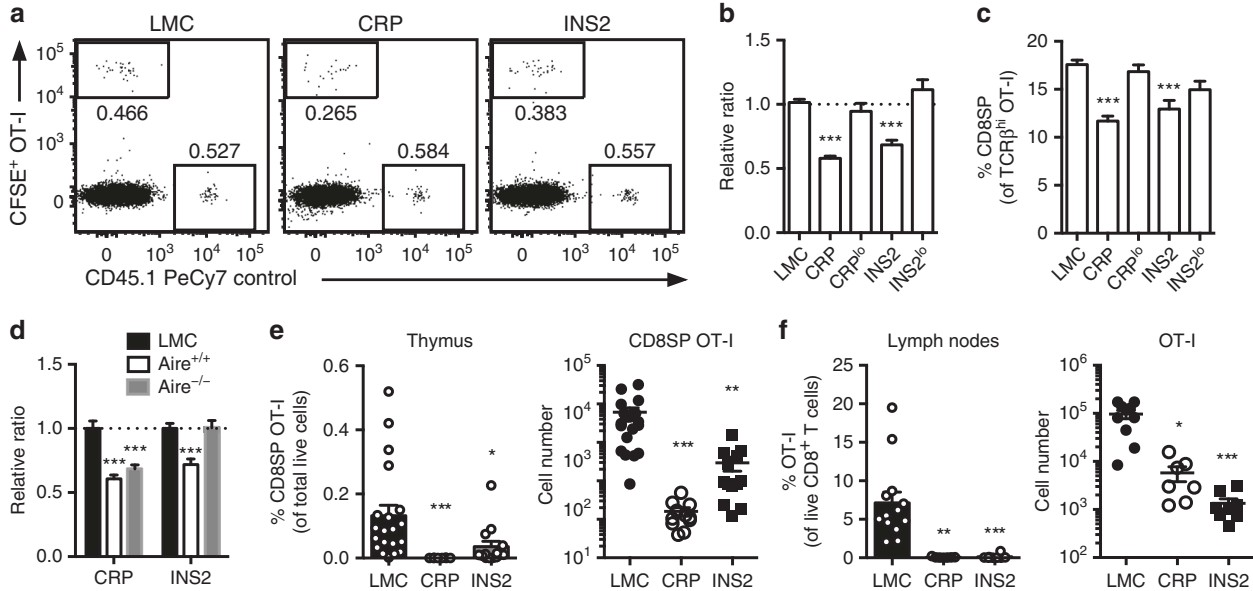

**Fig. 2 USA expression in the thymus of CRP and INS2 Tg mice support negative selection of antigen-specific MHC class I-restricted T cells.**
**a** Representative flow plot and **b** relative ratio of OT-I:control thymocytes 24 h after overlay on the indicated thymic slices. OT-I:WT control ratios were normalized to the littermate control (LMC) thymus (mean ± SEM, $n = 72$ LMC, 44 CRP, 12 CRP$^{lo}$, 26 INS2, 10 INS2$^{lo}$). **c** CD8$^+$ SP cells among live TCRβ$^{hi}$ OT-I cells 24 h after overlay on the indicated thymic slices. (mean ± SEM, $n = 100$ LMC, 48 CRP, 18 CRP$^{lo}$, 33 INS2, 10 INS2$^{lo}$). **d** Relative ratio of OT-I: control thymocytes 24 h after overlay on the indicated thymic slices (mean ± SEM, $n = 21$ LMC CRP, 17 Aire$^{+/+}$ CRP, 15 Aire$^{-/-}$ CRP, 12 INS2 LMC, 6 Aire$^{+/+}$ INS2, 12 Aire$^{-/-}$ INS2). **e**, **f** LMC, CRP and INS2 Tg mice (CD45.2) were irradiated and reconstituted with 1% OT-I (CD45.1) and 99% WT (CD45.1.2) bone marrow. Analysis of the proportion of mature TCRβ$^{hi}$ CD8$^+$ SP OT-I thymocytes among total live cells and absolute cell number of CD8SP OT-I in the thymus (**e**) (mean ± SEM, $n = 20$ LMC, 11 CRP, 13 INS2, from 3 independent experiments) and proportion of OT-I T cells among live CD8$^+$ T cells and absolute OT-I cell number in the peripheral lymph nodes (**f**) of low-density bone marrow chimeras 6–8 weeks post-reconstitution (mean ± SEM, $n = 12$ LMC, 7 CRP, 8 INS2, from 2 independent experiments). All the $p$-values indicated are calculated by Kruskal–Wallis test followed by Dunn's post-hoc comparisons to LMC, two-sided. *$p < 0.05$, **$p < 0.01$, ***$p < 0.001$ relative to LMC. Source data are provided as a Source Data file.

selection (late DP)[24] or at the CD8$^+$ SP stage[25], respectively, a significant proportion of IELp are YFP$^+$ suggesting that IELp diversion may also occur at the post-selection DP, and, perhaps to a lesser extent, the SP stage.

Model antigen expression in CRP Tg mice is preferentially detected in the mTEC$^{lo}$ population, but a significant proportion of mTEC$^{hi}$ express USA in these mice, raising the question of which cells are responsible for negative selection and differentiation of autoreactive thymocytes into CD8αα IELp. In order to better understand the role of each subset, mTEC$^{lo}$ and mTEC$^{hi}$ from WT and CRP Tg mice were sorted and co-cultured with OT-I thymocytes. Interestingly, mTEC$^{lo}$ and mTEC$^{hi}$ cells isolated from CRP Tg mice both induce a reduction in the proportion of OT-I CD8$^+$ SP cells suggesting that both are able to induce negative selection (Supplementary Fig. 3d, e). However, the differentiation of CD4$^-$CD8$^-$ DN cells is higher when OT-I thymocytes encounter cognate antigen presented by mTEC$^{lo}$ than by mTEC$^{hi}$ (Supplementary Fig. 3d, f) suggesting that reduced levels of costimulatory molecules may favor CD8αα IELp differentiation in this model.

**mTEC$^{hi}$ preferentially induce T$_{reg}$ differentiation.** Aire-dependent expression of TRA and co-stimulation are known to be important for thymic T$_{reg}$ differentiation[9,10,26–28]. Thus, we sought to determine if Aire-dependent expression of the model antigen in the INS2 Tg mice preferentially supports antigen-specific T$_{reg}$ differentiation in vivo. To do so, CRP and INS2 Tg mice were irradiated and reconstituted with 1% MHC class II-restricted OVA-specific OT-II TCR Tg Rag-deficient (OT-II) and 99% WT bone marrow, and the proportion and phenotype of the

OT-II cells were analyzed after reconstitution. The proportion and number of mature OT-II cells in the thymus and the peripheral lymph nodes is not significantly changed in the CRP and INS2 Tg BMC as compared to the LMC (Fig. 4a, b and Supplementary Fig. 4a–d). However, a significant increase in the proportion and absolute number of CD25$^+$FOXP3$^+$ OT-II cells is observed in the thymus and the lymph nodes from the low-density INS2 Tg BMC specifically (Fig. 4c–e). To determine the role of hematopoietically-derived antigen presenting cells (APC) in T$_{reg}$ induction in this model, CRP and INS2 Tg mice were irradiated and reconstituted with 1% OT-II and 99% MHC class II$^{-/-}$ bone marrow. A large proportion of OT-II cells still differentiate into CD25$^+$Foxp3$^+$ T$_{reg}$ in INS2 Tg mice as compared to LMC when the majority of hematopoietic cells do not express MHC class II (Supplementary Fig. 4e, f), suggesting that antigen presentation by mTEC$^{hi}$ is sufficient to induce OT-II diversion into CD25$^+$Foxp3$^+$ T$_{reg}$ in this model.

**TRA expression profiles impact polyclonal T cell fate.** We expanded our study to assess the fate of T cells in the polyclonal population specific for model antigen expressed by different mTEC populations. We observe differences in the number of tetramer positive cells between the LMC of the different USA-expressing mouse strains, so the results are normalized to the LMC in each line in order to be able to compare the different Tg mice. Consistent with our results using MHC class I-restricted TCR Tg models, a lower number of H-2K$^b$-OVA-specific CD8$^+$ T cells are found in the peripheral lymphoid organs from naïve CRP and INS2 Tg mice as compared to their respective LMC, suggesting partial clonal deletion of OVA-specific MHC class I

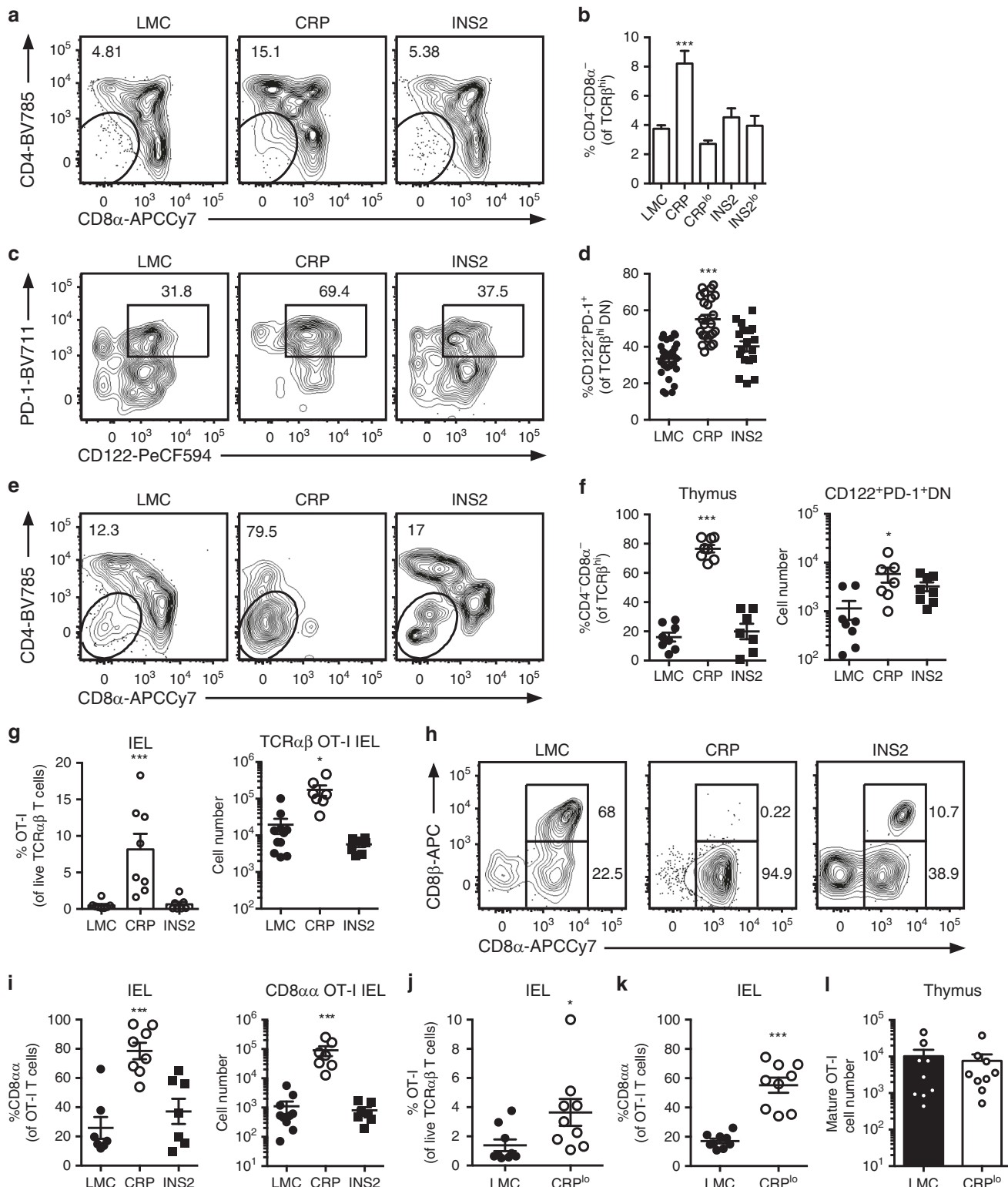

restricted polyclonal T cells (Fig. 5a, b and Supplementary Fig. 5a). Furthermore, naïve CRP Tg mice have a significant accumulation of H-2K$^b$-OVA-specific TCRαβ IEL in the small intestine as well as a higher proportion of TCRαβ$^+$ CD8αα IEL (Fig. 5c–e). For MHC class II-restricted model antigen-specific cells, we observe that the relative number of I-A$^b$-gp66-specific CD4$^+$ T cells is lower in both naïve CRP and INS2 Tg mice as compared to their respective LMC counterparts (Fig. 5f, g and Supplementary Fig. 5b), while only the INS2 Tg mice have an

increased proportion of model antigen-specific CD25$^+$Foxp3$^+$ T$_{reg}$ relative to LMC (Fig. 5h, i). However, the relative absolute number of gp66-specific CD25$^+$Foxp3$^+$ T$_{reg}$ is not increased in INS2 Tg mice as compared to LMC (Fig. 5i).

To determine the consequence of the thymic selection bias in immune responses, CRP and INS2 Tg mice were infected with lymphocytic choriomeningitis virus engineered to express OVA (LCMV-OVA[29]). Subsequently, the impact of model antigen expression in different mTEC compartments on USA-specific

**Fig. 3 mTEC$^{lo}$-biased expression of *USA* supports MHC class I-restricted TCRαβ$^+$ CD8αα intraepithelial lymphocyte differentiation. a** Representative flow plots of CD4 and CD8 on TCRβ$^{hi}$ OT-I thymocytes, and **b** proportion of CD4$^-$CD8$^-$ cells among live TCRβ$^{hi}$ OT-I thymocytes 24 h after overlay on the indicated thymic slices (mean ± SEM, n = 70 LMC, 30 CRP, 12 CRP$^{lo}$, 27 INS2, 12 INS2$^{lo}$). **c** Representative flow plot of PD-1 and CD122 and **d** proportion of CD122$^+$PD-1$^+$ cells among CD4$^-$CD8$^-$TCRβ$^{hi}$ OT-I T thymocytes 24 h after overlay on the indicated thymic slices (mean ± SEM, n = 30 LMC, 24 CRP, 18 INS2). **e–l** Littermate control (LMC), CRP, CRP$^{lo}$ and INS2 Tg mice (CD45.2) were irradiated and reconstituted with 1% OT-I (CD45.1) and 99% WT (CD45.1.2) bone marrow. OT-I cells in the indicated hosts were analyzed. **e** Representative flow plot of CD4 and CD8 expression on TCRβ$^{hi}$ OT-I thymocytes. **f** Proportion of CD4$^-$CD8$^-$ cells within the TCRβ$^{hi}$ OT-I thymocyte population and absolute number of PD1$^+$CD122$^+$TCRβ$^{hi}$ CD4$^-$CD8$^-$ OT-I thymocytes (mean ± SEM, left panel: n = 8 LMC, 8 CRP, 7 INS2, right panel: n = 8 LMC, 7 CRP, 8 INS2). **g** Proportion and cell number of OT-I T cells of TCRαβ intraepithelial lymphocytes (IEL) (mean ± SEM, left panel: n = 8 LMC, 8 CRP, 7 INS2, right panel: n = 10 LMC, 8 CRP, 8 INS2). **h** Representative flow plot of CD8α and CD8β staining on the OT-I T cells. **i** The proportion and cell number of TCRαβ$^+$ CD8αα IEL among the OT-I T cells (mean ± SEM, left panel: n = 8 LMC, 8 CRP, 7 INS2, right panel: n = 10 LMC, 8 CRP, 8 INS2). **j** Proportion of OT-I T cells within the live TCRαβ$^+$ IEL. **k** Proportion of TCRαβ$^+$ CD8αα of the OT-I T cells. **l** Proportion of mature TCRβ$^{hi}$ CD8$^+$ SP OT-I thymocytes among total live cells in the thymus (mean ± SEM, n = 9 mice per group). Data are from at least two independent experiments. The p-values for panels **b**, **d**, **f**, **g** and **i** were calculated by Kruskal–Wallis test followed by Dunn's post-hoc comparisons to LMC, two-sided. For panels **j–l** a two-tailed Mann-Whitney U test was performed. *p < 0.05, ***p < 0.001 relative to LMC. Source data are provided as a Source Data file.

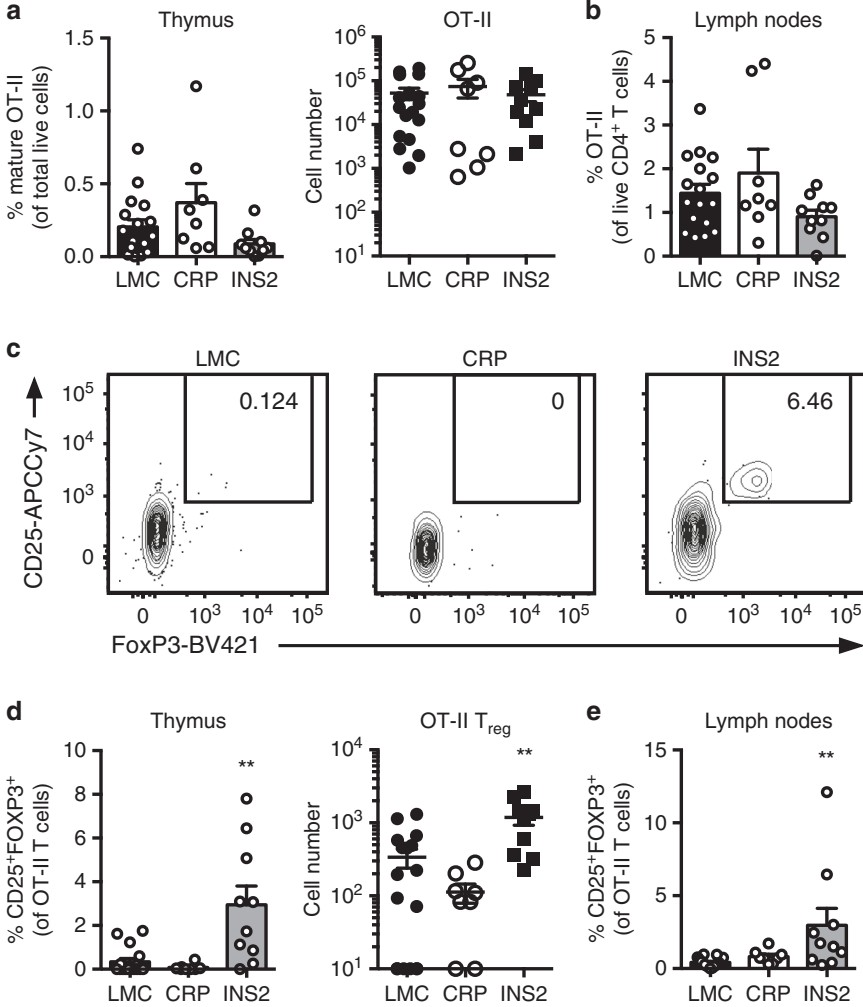

**Fig. 4 mTEC$^{hi}$-restricted expression of *USA* preferentially induces T$_{reg}$ differentiation.** Littermate control (LMC), CRP, and INS2 Tg mice (CD45.2, Thy1.2) were irradiated and reconstituted with 1% OT-II (CD45.2, Thy1.1) and 99% WT (CD45.1, Thy1.2) bone marrow. The hematopoietic chimeras were analyzed 6-8 weeks post-irradiation. **a** Proportion of mature TCRβ$^{hi}$ CD4$^+$ SP OT-II thymocytes among total live cells and absolute number of live CD4$^+$ SP OT-II thymocytes. **b** Proportion of OT-II cells within the CD4$^+$ T cell population in the peripheral lymph nodes. **c** Representative flow plots of CD25 and FoxP3 expression on OT-II T cells. **d** Proportion and absolute cell number of CD25$^+$Foxp3$^+$ OT-II T cells in the thymus. **e** Proportion of CD25$^+$Foxp3$^+$ OT-II T cells in the lymph nodes. (mean ± SEM, n = 17 LMC, 8 CRP, 10 INS2, from three independent experiments). All p-values indicated are calculated by Kruskal-Wallis test followed by Dunn's post-hoc comparisons to LMC, two-sided. **p < 0.01 relative to LMC. Source data are provided as a Source Data file.

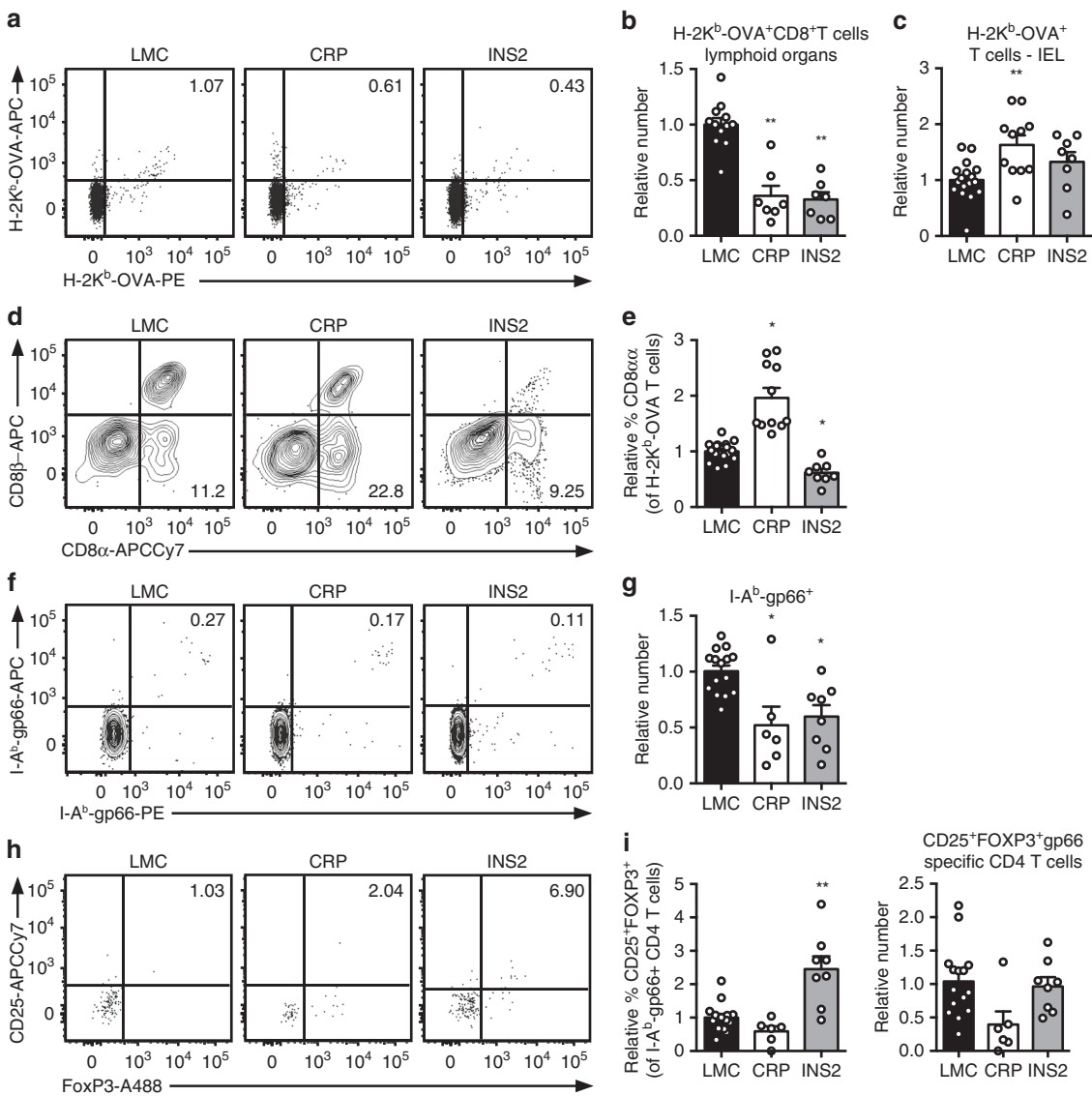

**Fig. 5 TRA expression by distinct mTEC compartments impacts the mode of tolerance induction in the polyclonal T cell repertoire in naïve mice.**
**a** Representative flow plots of H-2K$^b$-OVA tetramer on pooled peripheral lymph nodes and spleen cells from naïve mice after enrichment, gated on live TCRαβ$^+$ CD8$^+$ T cells. Relative number of H-2K$^b$-OVA-specific TCRαβ$^+$ CD8$^+$ T cells in the peripheral secondary lymphoid organs (mean ± SEM, $n = 12$ LMC, 7 CRP, 7 INS2) (**b**) and intraepithelial lymphocytes (IEL) (mean ± SEM, $n = 17$ LMC, 11 CRP, 8 INS2) (**c**) from CRP and INS2 Tg mice as compared to their respective littermate controls (LMC). **d** Representative flow plot of CD8α and CD8β and **e** relative proportion of TCRαβ$^+$ CD8αα within the H-2K$^b$-OVA tetramer double positive TCRαβ IEL of CRP and INS2 Tg mice as compared to their respective LMC (mean ± SEM, $n = 17$ LMC, 11 CRP, 8 INS2).
**f** Representative flow plots of I-A$^b$-gp66 tetramer on pooled peripheral lymph nodes and spleen cells from naïve mice, gated on live TCRαβ$^+$ CD4$^+$ T cells.
**g** Relative number of I-A$^b$-gp66 tetramer DP CD4$^+$ T cells in the peripheral lymph nodes and the spleen of naïve mice. **h** Representative flow plots of CD25 and FoxP3, gated on live I-A$^b$-gp66 tetramer$^+$ CD4$^+$ T cells. **i** Relative proportion of CD25$^+$FoxP3$^+$ T$_{reg}$ of gp66-specific CD4$^+$ T cells and relative number of gp66-specific CD4$^+$CD25$^+$FoxP3$^+$ T$_{reg}$ from CRP and INS2 peripheral lymph nodes and spleen as compared to their respective LMC. Absolute numbers were normalized to LMC for each Tg and in each experiment for the relative number calculation (**g** and **i**, mean ± SEM, $n = 15$ LMC, 6 CRP, 8 INS2). Data are pooled from at least two independent experiments. All the p values indicated are calculated by Kruskal–Wallis test followed by Dunn's post-hoc comparisons to LMC. *$p < 0.05$, **$p < 0.01$ relative to LMC. Source data are provided as a Source Data file.

MHC class I- and class II- restricted T cells was assessed. In accordance with the number of OVA-specific CD8$^+$ T cells in naïve CRP and INS2 Tg mice, a lower proportion and absolute number of H-2K$^b$-OVA-specific CD8$^+$ T cells is observed in the *USA* expressing mice as compared to their respective LMC 8 days post LCMV-OVA infection (Fig. 6a, b and Supplementary Fig. 5c). Moreover, a significant decrease in the proportion of LCMV gp33-specific CD8$^+$ T cells is observed in the CRP and INS2 mice (Fig. 6c, d). No reduction in the LCMV NP396-specific response is observed in any of the *USA*-expressing mice

as compared to LMC, confirming that the decreased immune response is specific to the epitopes present in the USA construct (Supplementary Fig. 5d, e). In addition, a lower proportion and absolute number of LCMV gp66-specific CD4$^+$ T cells is observed in both CRP and INS2 Tg mice after LCMV infection (Fig. 6e, f), but only the INS2 Tg mice show an increase in gp66-specific CD25$^+$Foxp3$^+$ T$_{reg}$ (Fig. 6g, h). Surprisingly, while the LCMV-OVA infected CRP$^{lo}$ Tg mice do not display any reduction in the gp66-specific CD4$^+$ T cells response, the INS2$^{lo}$ Tg mice have reduced gp66-specific CD4$^+$ T cells of

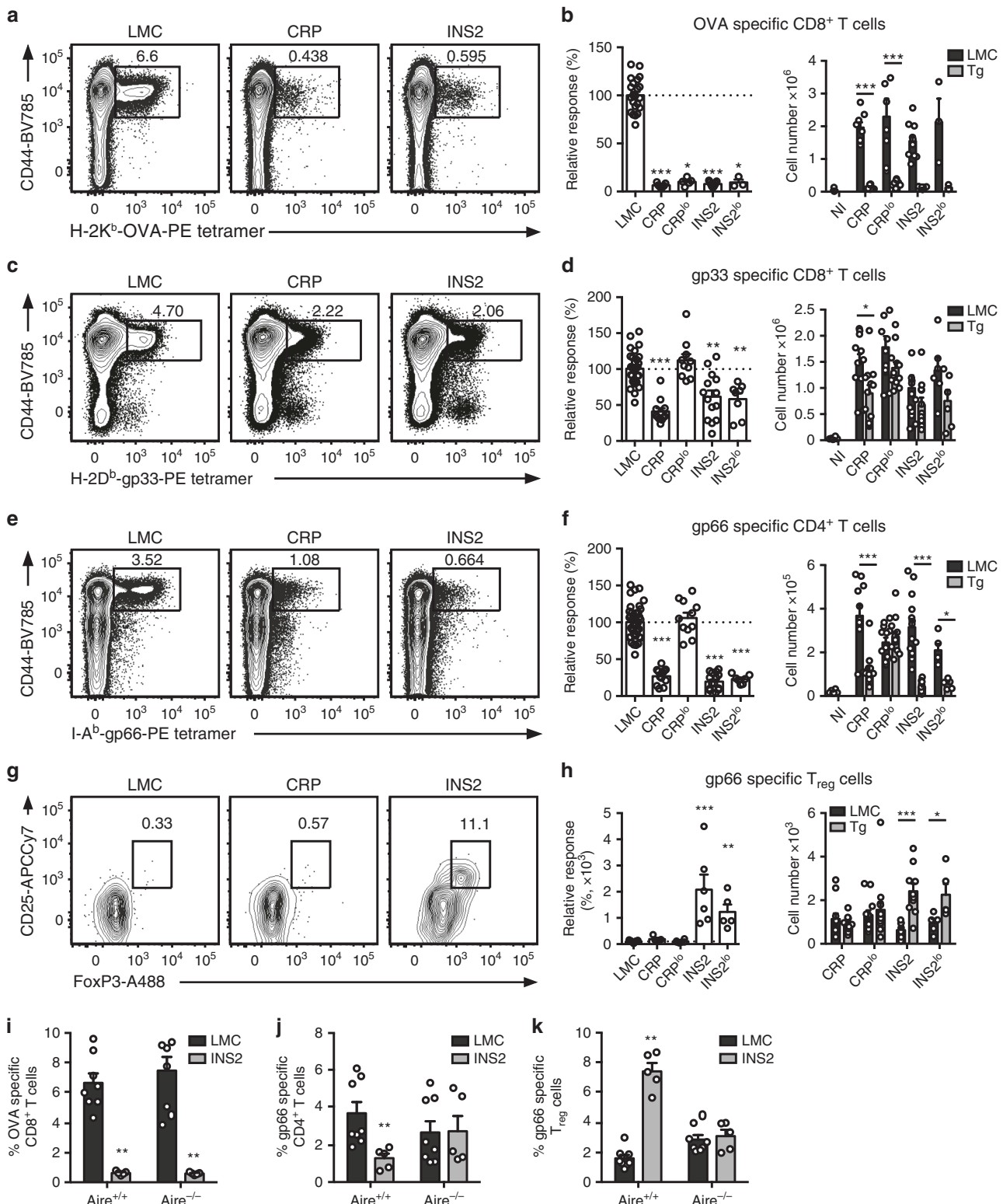

which a high proportion are CD25⁺Foxp3⁺ T$_{reg}$ (Fig. 6e–h). Interestingly, when Aire$^{−/−}$ INS2 Tg mice, in which the transgene expression is lost in the thymus (Fig. 1b), are infected with LCMV-OVA, we still observe a significant reduction in the proportion of OVA-specific CD8⁺ T cells at day 8 post-infection, suggesting that Aire-independent peripheral expression of OVA is sufficient to dampen the OVA-specific effector CD8⁺ T cells response (Fig. 6i). However, analysis of the gp66-specific CD4⁺ T cell response suggests that peripheral

transgene expression is not sufficient to decrease the effector response nor to increase the proportion of gp66-specific T$_{reg}$, suggesting that a central tolerance mechanism requiring expression of the TRA in the thymus is dominant (Fig. 6j, k). In sum, the data revealed by the tetramer staining of polyclonal T cells from naïve and infected mice confirm that mTEC$^{lo}$ biased and mTEC$^{hi}$ restricted expression of model antigen induce distinct autoreactive T cell fates in the context of a polyclonal TCR repertoire in these models.

**Fig. 6 TRA expression by distinct mTEC compartments impacts the mode of tolerance induction in the polyclonal T cell repertoire in LCMV infected mice.** Mice were infected with LCMV-OVA and the antigen-specific response was measured in the spleen 8 days post-infection (NI, not infected mice) **a, c** Representative CD44 and H-2K$^b$-OVA or H-2D$^b$-gp33 tetramer staining on CD8$^+$ T cells from littermate controls (LMC), CRP, and INS2 Tg mice. Relative response and absolute number of OVA- (**b**) and gp33- (**d**) specific CD8$^+$ T cell response in USA expressing mice as compared to LMC. **e** Representative CD44 and I-A$^b$-gp66 tetramer staining on CD4$^+$ T cells from LMC, CRP, and INS2 Tg mice. **f** Relative and absolute gp66-specific response in USA expressing mice. **g** Representative flow plot of CD25 and FoxP3, gated on live I-A$^b$-gp66-specific CD4$^+$ T cells. **h** Relative proportion of CD25$^+$FoxP3$^+$ T$_{reg}$ among gp66-specific CD4$^+$ T cells and absolute number of gp66-specific CD4$^+$CD25$^+$FoxP3$^+$ T$_{reg}$ (mean ± SEM, **b** left panel $n = 22$ LMC, 7 CRP, 5 CRP$^{lo}$, 9 INS2, 3 INS2$^{lo}$; **b** right panel $n = 3$ NI, 8 CRP LMC, 10 CRP Tg, 9 CRP$^{lo}$ LMC, 9 CRP$^{lo}$ Tg, 10 INS2 LMC, 9 INS2 Tg, 3 INS2$^{lo}$ LMC, 3 INS2$^{lo}$ Tg; **d** and **f** left panel $n = 32$ LMC, 12 CRP, 12 CRP$^{lo}$, 14 INS2, 8 INS2$^{lo}$, **d** right panel $n = 6$ NI, 11 CRP LMC, 12 CRP Tg, 11 CRP$^{lo}$ LMC, 12 CRP$^{lo}$ Tg, 13 INS2 LMC, 11 INS2 Tg, 6 INS2$^{lo}$ LMC, 6 INS2$^{lo}$ Tg, **f** right panel $n = 6$ NI, 11 CRP LMC, 13 CRP Tg, 12 CRP$^{lo}$ LMC, 13 CRP$^{lo}$ Tg, 14 INS2 LMC, 12 INS2 Tg, 5 INS2$^{lo}$ LMC, 5 INS2$^{lo}$ Tg; **h** left panel $n = 21$ LMC, 5 CRP, 6 CRP$^{lo}$, 6 INS2, 5 INS2$^{lo}$, **h** right panel $n = 11$ CRP LMC, 13 CRP Tg, 12 CRP$^{lo}$ LMC, 13 CRP$^{lo}$ Tg, 9 INS2 LMC, 10 INS2 Tg, 5 INS2$^{lo}$ LMC, 5 INS2$^{lo}$ Tg). Proportion of OVA-specific CD8$^+$ T cells (**i**), gp66-specific CD4$^+$ T cells (**j**) and gp66-specific T$_{reg}$ (**k**) in Aire$^{+/+}$ and Aire$^{-/-}$ LMC and INS2 Tg mice (mean ± SEM, $n = 8$ Aire$^{+/+}$ LMC, 5 Aire$^{+/+}$ INS2, 8 Aire$^{-/-}$ LMC, 5 Aire$^{-/-}$ INS2). The relative number of tetramer$^+$ cells was obtained by dividing the number of tetramer$^+$ cells of each mouse by the mean number of the corresponding LMC mice in each experiment. Data from three independent experiments. The $p$-values for the left panels of **b**, **d**, **f**, **h** were calculated by Kruskal–Wallis test followed by Dunn's post-hoc comparisons to LMC, two-sided. For the right panels in **b**, **d**, **f**, **h** and panels **i**–**k** a two-tailed Mann–Whitney U test was performed. *$p < 0.05$, **$p < 0.01$, ***$p < 0.001$, relative to LMC. Source data are provided as a Source Data file.

**Differential TRA expression in mTEC impacts T cell function.** TCRαβ$^+$ CD8αα IELs play context-dependent immunoprotective and pathologic roles in intestinal homeostasis[30,31]. To determine if the higher proportion of OVA-specific TCRαβ$^+$ CD8αα IEL in CRP Tg mice have functional consequences, low-density OT-I BMC were infected by oral gavage with *Listeria monocytogenes* engineered to express OVA (Lm-OVA), and the number of bacteria in the small intestine and mesenteric lymph nodes were enumerated three days later, prior to significant influence from conventional TCRαβ$^+$ populations[32,33]. Interestingly, CRP OT-I BMC have a higher number of bacteria in the mesenteric lymph nodes and the small intestine following Lm-OVA infection as compared to LMC and INS2 OT-I BMC (Fig. 7a). In addition, the increased number of bacteria is observed with Lm-OVA, but not Lm-WT, bacterial infections (Fig. 7b) consistent with the idea that TCRαβ$^+$ CD8αα IELs require antigen-specific TCR stimulation for activation[34].

T$_{reg}$ are known to influence tumor control[35]. Thus, we evaluated growth of B16-OVA tumors in LMC, CRP, and INS2 Tg mice. The kinetics of tumor growth is similar in LMC and CRP Tg mice inoculated with B16-OVA cells (Fig. 7c). However, B16-OVA tumor growth is accelerated in INS2 mice as compared to LMC. The exacerbated tumor growth in INS2 Tg mice is both Aire and OVA antigen dependent (Fig. 7c, d). In addition, depletion of CD25$^+$ T$_{reg}$ in INS2 mice restores control of tumor growth to similar levels as in LMC (Fig. 7e). Together, these data suggest an important role of thymic derived OVA-specific T$_{reg}$ in the loss of control of tumor growth observed in the INS2 Tg mice.

## Discussion

Recent single-cell RNA sequencing studies of murine TECs have bolstered our understanding of the diversity of mTEC populations[12,13]. These studies reinforce the idea that distinct mTEC populations express different subsets of TRA, but whether they play unique roles directing autoreactive T cell fates is less clear[12–14,21,36,37]. In the present study, we generate new Tg mouse models with mTEC$^{lo}$-biased or mTEC$^{hi}$-restricted expression of model antigens that lead to differences in the quality of tolerance induction. These models not only allow us to confirm an important role of Aire-dependent TRA expression by mTEC$^{hi}$ in T$_{reg}$ differentiation, but also uncover a potential role for mTEC$^{lo}$ biased antigen expression in the diversion of autoreactive T cells into TCRαβ$^+$ CD8αα IELs with physiologically important differences in modulating the immune response. Thus, we show that the mechanism of tolerance may not only be based on TCR affinity for self-peptide and the number of antigen expressing

cells in the thymus, but may also be influenced by the mTEC subset expressing the self-antigen.

In our model, antigen expression predominantly in the mTEC$^{lo}$ compartment supports antigen-specific thymocyte diversion toward the TCRαβ$^+$ CD8αα IEL lineage. TCRαβ$^+$ CD8αα IELs are a heterogenous cell population that arise from distinct developmental progenitors[31,38]. Two TCRαβ$^+$ CD8αα IEL populations develop in the thymus and differ in their self-reactivity[39]. The IELp population that expresses PD-1 is enriched in self-reactive cells that are induced after agonist selection[31,39]. Increases in IELp have been observed in the absence of co-stimulatory molecule expression by thymocytes[11] suggesting that autoreactive thymocyte interactions with APCs bearing low expression of costimulatory molecules are required to induce diversion toward this lineage. Indeed, diversion of autoreactive thymocytes into TCRαβ$^+$ CD8αα IELp is generally thought to occur at the DP stage or during the DN-to-DP transition after agonist signal[4,39–46], and DP thymocytes are largely located within the cortex limiting their interaction with mTEC. Nevertheless, in our study, mTEC$^{lo}$ biased expression of model antigens induces differentiation of antigen-specific thymocytes into IELp and generates a high proportion of antigen-specific TCRαβ$^+$ CD8αα IEL. In addition, the use of reporter mice in which developmentally distinct thymocyte subsets express YFP support the idea that the majority of IELp arise from pre-selection DP thymocytes. Although we cannot rule out that reporter expression is upregulated after differentiation, we also provide evidence that a portion of IELp possibly differentiate in response to agonist signals at the post-selection DP or even CD8$^+$ SP stage supporting the idea that differentiation signals may also be provided in the medulla following the recognition of TRA expressed by mTEC$^{lo}$. In further support of this notion, induction of a high proportion of antigen-specific TCRβ$^+$ DN thymocytes has been observed in the RIP-mOVA model in which OVA is highly expressed in both mTEC$^{lo}$ and mTEC$^{hi}$ cells[28,44,47]. Moreover, in vitro culture of OT-I thymocytes with sorted mTEC$^{lo}$ isolated from CRP Tg mice leads to OT-I differentiation into TCRβ$^{hi}$CD4$^-$CD8$^-$ DN cells confirming that these cells are able to divert autoreactive thymocytes into CD8αα IELp. In fact, while both mTEC$^{lo}$ and mTEC$^{hi}$ induce similar deletion of OT-I thymocytes, mTEC$^{lo}$ seem to be more efficient at inducing OT-I differentiation into TCRβ$^{hi}$CD4$^-$CD8$^-$ DN cells supporting the idea that absence of co-stimulation favors differentiation into CD8αα IELp. mTEC$^{lo}$ and mTEC$^{hi}$ cells from CRP Tg mice express similar levels of USA mRNA on a per cell basis suggesting that the level of TRA expression by individual cells is not a

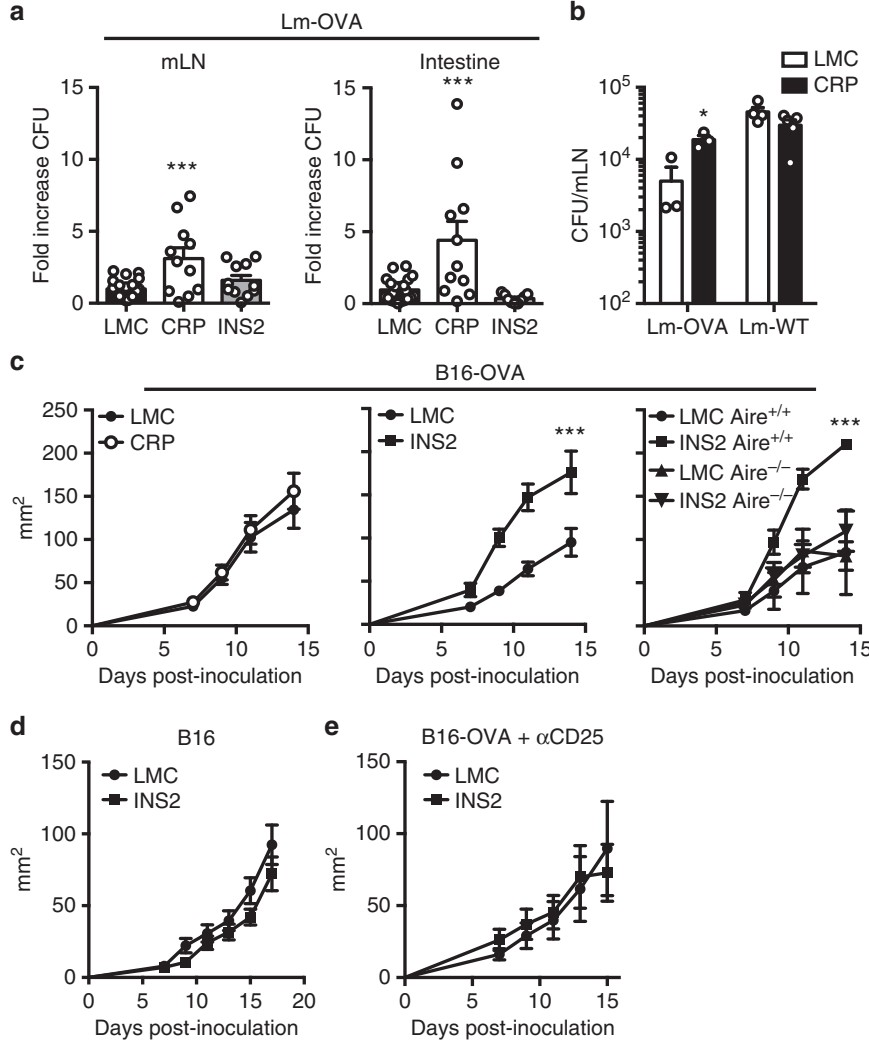

**Fig. 7 Biased expression of TRA in distinct mTEC subsets differentially impacts control of bacterial infection and tumor growth.** Low-density OT-I hematopoietic chimeras were infected by oral gavage with *Listeria monocytogenes* OVA (Lm-OVA) (**a, b**) or Lm-WT (**b**) and euthanized 3 days post-infection to determine the number of Lm bacteria in the mesenteric lymph nodes (mLN) and the small intestine. The results are reported as fold increase in colony forming units (CFU) as compared to littermate control (LMC) mice (mean ± SEM, $n = 20$ LMC, 10 CRP, 10 INS2 mice) (**a**) and in CFU in the mLNs from the LMC and CRP Tg mice (mean ± SEM, Lm-OVA, $n = 3$ per group, Lm-WT, $n = 4$ per group) (**b**). B16-OVA **c** (mean ± SEM, $n = 8$ CRP LMC, 10 CRP Tg, 10 INS2 LMC, 10 INS2 Tg, 2 LMC Aire$^{+/+}$, 3 Aire$^{+/+}$ INS2, 5 Aire$^{-/-}$ LMC, 4 Aire$^{-/-}$ INS2) or B16 **d** (mean ± SEM, $n = 7$ per group) cells were inoculated subcutaneously, and tumor growth was followed every 2 days by measuring tumor diameter and calculating the tumor area in the indicated mice. **e** INS2 Tg mice were injected with anti-CD25 depleting antibody 4 days and 1 day before B16-OVA cell inoculation, and tumor growth was measured every 2 days starting on day 7 post-inoculation (mean ± SEM, $n = 10$ per group). Data are pooled from two independent experiments. Statistical analyses were performed by Kruskal–Wallis test followed by Dunn's post-hoc comparisons to LMC, two-sided **a**, a two-tailed Mann–Whitney U test **b** and linear mixed-effects modeling **c**–**e**. *$p < 0.05$, ***$p < 0.001$ relative to LMC. Source data are provided as a Source Data file.

determining factor for CD8αα IEL differentiation. Moreover, lower global model antigen expression in CRP$^{lo}$ Tg mice also induces significant differentiation of autoreactive cells into CD8αα IEL pointing to a role for the type of cells expressing the antigen rather than antigen level in these models.

TCRαβ$^+$ CD8αα IEL play an important role in intestinal homeostasis. In this study, we observe that mice with a higher number of OVA-specific TCRαβ$^+$ CD8αα IEL are less able to control model antigen-expressing bacteria growth and spread after oral challenge. Following antigen recognition, model antigen-specific TCRαβ$^+$ CD8αα IELs could produce immuno-modulatory cytokines leading to a decreased defense against pathogens. It is also conceivable that the role of these cells is to reduce inflammation at a barrier site or to specifically inhibit an autoimmune response in order to maintain tolerance against self

and commensal microorganisms[34,48,49]. However, the mechanism by which the higher number of OVA-specific TCRαβ$^+$ CD8αα IELs leads to a greater Lm-OVA burden remains to be determined in this context. In addition, the role of Aire-independent TRA-specific TCRαβ$^+$ CD8αα IEL in maintaining intestinal homeostasis requires further study. While a significant number of intestinal TRA are expressed in the thymus, they are not significantly enriched in the mTEC$^{lo}$ compartment[16]. It is possible that a subset of TCRαβ$^+$ CD8αα IEL selected on TRA expressed by mTEC$^{lo}$ simply serves to expand the TCR repertoire of this population or that this subset plays a unique role in intestinal homeostasis.

Considering that autoreactive T cell fate is determined by TCR signal strength, which is, itself, influenced by TCR affinity and avidity, it is not surprising that, in addition to TCR sequence

itself[26,50], the number of MHC and costimulatory molecules expressed by the antigen presenting cell influences the outcome of this interaction. Only CRP Tg mice express *USA* in mTEC[lo] at detectable levels. Thus, if we assume that autoreactive T cell diversion into IELp is induced in the absence of co-stimulation, it makes sense that CRP, and not INS2 Tg mice can support their differentiation, and one might predict that CD80/86-deficient INS2 Tg mice would also support CD8αα IEL development. In contrast, it is well accepted that T[reg] differentiation requires co-stimulation and that Aire-dependent expression of model antigen in mTEC[hi] supports T[reg] differentiation[3,51]. However, both INS2 and CRP Tg mice express similar levels of *USA* in the mTEC[hi] subpopulation raising the question as to why autoreactive T cell diversion in T[reg] is enhanced only in INS2 Tg mice. It is possible that, in CRP Tg mice, model antigens are mainly expressed by a transitional mTEC[hi] population expressing lower levels of costi-mulatory molecules thus decreasing their capacity to induce T[reg] differentiation. In addition, it was been shown in some models that antigen transfer to or direct antigen expression by BM-APC was important for negative selection and T[reg] differentiation[28,52,53]. As these processes are dependent on Aire[52,54,55], it is possible to speculate that these mechanisms are less efficient for Aire-independent TRA, reducing their capacity to induce T[reg] differentiation. However, MHCII deletion on BM-APC did not abrogate T[reg] differentiation in INS2 Tg BMC suggesting that TRA presentation by BM-APC is not the main mechanism triggering T[reg] differentiation in our model and is consistent with other studies showing that MHCII deletion on BM-APC have only a mild impact on T[reg] generation[51,56]. It is also conceivable that although CRP Tg mice express model antigen in mTEC[hi], TRA expression in mTEC[lo] reduces the chance of interaction between autoreactive T cells and mature mTEC therefore limiting the diversion of T cells into T[reg]. Separately, it has been demonstrated that the mechanism of CD4[+] T cell tolerance is influenced by the number of self-antigen expressing cells in the thymus[8,57]. Using mice with more subtle differences in model antigen expression level (CRP vs CRP[lo] and INS2 vs INS2[lo] Tg mice), we suggest that overall levels of model antigen in the thymus has a large impact on the extent, but not necessarily the type, of the tolerance mechanisms induced in these Tg models. Instead, we observed, by single-cell RNA sequencing, that *Ins2* is expressed in very few mTEC[hi] cells, but its expression level on a per cell basis is higher than that of *Crp*, suggesting that T[reg] differentiation might require, in addition to high co-stimulatory molecule expression, relatively higher TRA expression on a per cell basis. Thus, according to our study, the pattern of antigen expression across different mTEC populations, and, potentially, the expression level of self-antigen on a per cell basis, may impact T[reg] differentiation.

In conclusion, mTEC[hi] restricted and mTEC[lo] biased model antigen expression induces distinct autoreactive T cells fates in these Tg models. As such, we suggest that differential expression of TRA among mTEC subsets may contribute to the type of T cell tolerance generated; it will be important to determine the extent to which these observations are generalizable to endogenous Aire-dependent and -independent TRA.

## Methods

**Mice**. C57BL/6 (Stock# 000664), B6.SJL (Stock# 002014), *Aire*[−/−] (B6.129S2-Airetm1.1Doi/J, Stock# 004743)[58], and β2 M[−/−] (B6.129P2-B2mtm1Unc/J, Stock# 002087)[59] mice were purchased from The Jackson Laboratory. OT-I Rag1-deficient mice were obtained through the NIAID Exchange Program (NIH:C57BL/6-Tg(OT-I)Rag1 < tm1Mom> Stock# 2334)[60]. OT-II Rag1-deficient mice were obtained from the National Institutes of Health Taconic repository (Stock# 11490)[61]. P14 TCR Tg mice[62] were provided by Dr. Martin Richer (McGill University, Montreal, Canada) and crossed onto a TCRα KO (The Jackson Laboratory, Stock# 002116)[63] back-ground. MHCII[−/−]CD1d[−/−] (B6.129PS2-H2 dlabl-Ea/J (The Jackson Laboratory,

Stock# 003584)[64] x B6.129S6-Del(3Cd1d2-Cd1d1)1Sbp/J (The Jackson Laboratory, Stock# 008881)[65], CD4-YFP (B6.Cg-Gt(ROSA)26Sortm3(CAG-EYFP)Hze/Jv (The Jackson Laboratory, Stock# 007903)[66] x B6.Cg-Tg(Cd4-cre)1Cwi/BfluJ (The Jackson Laboratory, Stock# 022071)[67], dLCK-YFP (B6.Cg-Gt(ROSA)26Sortm3(CAG-EYFP)Hze/Jv x B6.Cg-Tg(Lck-icre)3779Nik/J (The Jackson Laboratory, Stock# 012837)[68] and E8i-YFP (B6.Cg-Gt(ROSA)26Sortm3(CAG-EYFP)Hze/Jv x C57BL/6-Tg(Cd8a-cre)1Itan/J (The Jackson Laboratory, Stock# 008766)[69], were kindly provided by Dr Sylvie Lesage (Maisonneuve-Rosemont Hospital Research Center, Université de Montréal, Montreal, Canada). CRP, CRP[lo], INS2 and INS2[lo] Tg mice were generated via BAC Recombineering[70,71]. BAC clones RP23-51J21 and RP23-160P13 containing *Ins2* and *Crp*, respectively, near the middle of the cloned genomic region were procured from the BACPAC Resources Center. USA-T2A-eGFP was recombineered into each clone at the start codon of the gene of interest. The USA-T2A-eGFP knock-in was validated by sequencing. BAC DNA was microinjected in C57BL/6 embryos to generate founders by the transgenic core facility at the Institute for Research in Immunology and Cancer (Université de Montréal). Mice were maintained in a specific pathogen-free environment at the Maisonneuve-Rosemont Hospital Research Center under a 12 h/12 h light/dark cycle. Temperature was maintained at 22 °C with a relative humidity of 40%. Experimental control mice were co-housed. Euthanasia was performed by cervical dislocation or carbon dioxide inhalation. All animal protocols were approved by the local Animal Care Committee at the Maisonneuve-Rosemont Hospital in accordance with the Canadian Council on Animal Care guidelines.

**Virus and Cells**. MC57G and L929 fibroblasts were cultured in MEM containing 5% heat-inactivated fetal bovine serum (FBS). LCMV-OVA Armstrong was obtained from Juan C. de la Torre (The Scripps Research Institute, La Jolla, CA, USA[29]) and was subsequently propagated by infection of the L929 fibroblast cell line and harvesting of virus in the supernatant. LCMV titers were determined using MC57G fibroblasts[72]. B16F10 and B16-OVA cells were kindly provided by Alain Lamarre (INRS-Institut Armand-Frappier, Laval, Quebec, Canada) and cultured in DMEM supplemented with 10% FBS and 5 mg ml[−1] of G418 to select for OVA expression.

**Immune cell isolation**. Spleens and lymph nodes were dissociated using frosted glass slides, while single-cell suspensions of the thymus were prepared using a glass tissue homogenizer. Red blood cells in spleens and thymus were lysed before analysis with Ack lysis buffer (0.15 M NH$_4$Cl, 10 mM KHCO$_3$, 0.1 mM Na$_2$EDTA). To isolate IEL from the small intestine, the connective tissue, fat, Peyer's patches and feces were removed. One cm intestine pieces were incubated in RPMI media supplemented with 3% FBS, 5 mM EDTA and 0.145 mg ml[−1] DL-Dithiotreitol (Sigma-Aldrich) for 20 min at 37 °C with agitation. The IEL were then isolated by shaking the small intestine pieces in RPMI media with 2 mM EDTA followed by a 30% Percoll (GE Healthcare) gradient of the released cells. For isolation of the thymic epithelial cells, the thymus was first cut in small pieces and pipetted up and down with a wide bore 1000 μl tip until cell release was no longer detectable. The remaining tissue was digested in RPMI media supplemented with 10% FBS, 10 mM HEPES, 0.25 mg ml[−1] papain, 0.25 mg ml[−1] collagenase D, 0.1 mg ml[−1] DNase for 15 min at 37 °C. The TEC were then enriched using EpCAM magnetic microbeads (Miltenyi Biotec) and sorted using a BD FACS Aria III with an 85 μM nozzel.

**Infections**. Six to twelve week-old female mice were infected with $2 \times 10^5$ focus forming units (FFU) of LCMV-OVA Armstrong intraperitoneally (i.p.) and spleens were harvested 8 days post-infection for flow cytometry analysis. *Listeria mono-cytogenes* (Lm) infections were performed by gavage of female mice with $1 \times 10^9$ colony forming units (CFU) of Lm-WT or Lm-OVA bacteria. The mesenteric lymph nodes and the small intestine were harvested 3 days post-infection and homogenized in distilled water plus 0.5% Nonidet P-40 (Bio Basic). Fold serial dilutions were then plated onto brain heart infusion (VWR) agar plates containing 200 mg ml[−1] streptomycin (Bio Basic). Plates were incubated at 37 °C for 24 h, and colonies were enumerated.

**Bone marrow chimeras**. Six to twelve week-old male or female donor mice were injected i.p. with 100 μg InVivoMAb anti-mouse Thy1 (BioXcell) at day −2 and −1 prior to harvesting the bone marrow cells. Six to twelve week-old sex matched male or female recipient mice were irradiated at 12 Gy and reconstituted by injecting $5 \times 10^6$ T cell depleted bone marrow cells from the donor intrave-nously (i.v.). The recipient mice were then injected i.p. with 100 μg InVivoMAb anti-mouse Thy1 (BioXcell) 1 and 7 days after bone marrow cells injection. BMC were analyzed 6-8 weeks post-irradiation.

**Adoptive cell transfer**. Lymph nodes of six to twelve week-old male or female OT-I and B6.SJL mice were harvested, cells were mixed at a 1:1 ratio, and $4 \times 10^6$ cells were injected i.v. in six to twelve week-old sex matched male or female *USA*-expressing or LMC mice. Six to eight weeks later, the proportion of TCRαβ[+] CD8αα IEL among the OT-I T cells isolated from the small intestine was analyzed by flow cytometry.

**Thymic slices**. The thymus of five to eight week-old male or female USA-expressing or LMC mice were harvested, embedded in 4% low melt agarose and cut in 500 μm slices[73]. CFSE labeled five to twelve week-old sex matched male or female OT-I and B6.SJL thymocytes were mixed in a 1:1 ratio and a total of $2 \times 10^6$ cells were overlaid on top of the thymic slices and incubated at 37 °C for 24 h. Alternatively, $1 \times 10^6$ mature CFSE labeled OT-I T cells isolated from the lymph nodes were overlaid on top of the thymic slices and incubated at 37 °C for 72 h. The thymic slices were dissociated and analyzed by flow cytometry.

**Flow cytometry**. Flow cytometry analysis of mouse surface antigens was performed with the following Abs: Anti-CD3 (145-2C11, Cat#100328, 1:100), -CD4 (GK1.5, Cat#100422, 1:1600; RM4-5, Cat#100552, 1:400), -CD8α (53-6.7, Cat#100714, 1:400), -CD8β (H35-17-2, Cat#126616, 1:400), -CD11c (N418, Cat#117308, 1:1600), -CD19 (6D5, Cat#115505, 1:800) -CD25 (PC61, Cat#102026, 1:200), -CD44 (IM7, Cat#103059, 1:200), -CD45 (30-F11, Cat#103125, 1:400), -CD45.1 (A20, Cat#110730, 1:400), -CD45.2 (104, Cat#109822, 1:100), -CD80 (16-10A1, Cat#104722, 1:400), -CD90.1 (OX-7, Cat#202528, 1:400), -CD122 (TM-b1, Cat#123218, 1:400), -CD326 (G8.8, Cat#118218, 1:200), -F4/80 (BM8, Cat#123118, 1:200) -Ly-51 (6C3, Cat#108312, 1:200), -PD-1 (CD279) (29 F.1A12, Cat#135231, 1:200), -I-A/I-E (M5/114.15.2, Cat#107630, 1:1600), -TCRβ (H57-597, Cat#109206, 109224, 1:100), -TCR Vα2 (B20.1, Cat#127808, 1:1600), -TCR Vβ5 (MR9-4, Cat#139508, 1:200) (BioLegend) and biotinylated Ulex Europaeus Agglutinin I (UEA 1) (Vector Laboratories). Staining was performed for 20 min at 4 °C. The H-2K$^b$-OVA$_{257-264}$, H-2D$^b$-GP$_{33-41}$, H-2D$^b$-NP$_{396-404}$ and I-A$^b$GP$_{66-77}$ biotinylated monomers were obtained through the NIH Tetramer Core Facility and tetramers were generated using extravidin-PE (Thermo Fisher Scientific). Tetramer staining was performed at 37 °C for 15–30 min. Intranuclear staining for FoxP3 (150D/E4, Cat#53-4774-42, 1:50; MF23, Cat#562996, 1:100) (Thermo Fisher Scientific) was performed using fixation/permeabilization buffer (Thermo Fisher Scientific), according to the manufacturer's instructions. As needed, Fc receptors were blocked using anti-CD16/32 (93, Cat#101321, 1:50) antibody and the cells were stained with Zombie Aqua fixable viability dye (BioLegend) for 15 min at room temperature. Flow cytometry analyses were performed on a BDLSR Fortessa X20 or BDLSRII flow cytometer (BD Biosciences) and data were analyzed using the FlowJo software (BDBiosciences).

**qPCR**. For mRNA quantification, thymic B cells (CD19$^+$), DC (CD11c$^+$), macrophages (F4/80$^+$), mTEC$^{lo}$ (EpCAM$^+$CD45-UEA-1$^+$Ly51$^-$CD80$^{lo}$MHC-II$^{lo}$) and mTEC$^{hi}$ (EpCAM$^+$CD45$^-$UEA-1$^+$Ly51$^-$CD80$^{hi}$MHC-II$^{hi}$) were sorted directly into TRIzol LS (ThermoFisher Scientific) from five to ten week-old male or female mice. Spleens were dissociated using frosted glass slides and a red blood cell lysis was performed. Livers were cut in small pieces and digested for 15 min at 37 °C in RPMI media with 1 mg ml$^{-1}$ collagenase D. The liver pieces were then passed through a 100 μm cell strainer and a red blood cell lysis was performed. For spleen and liver samples, $1 \times 10^6$ cells were pelleted and resuspended in 1 ml TRIzol. Pancreas were injected with HBSS containing 10% FBS, 0,1 mg ml$^{-1}$ DNAse I and 1 mg ml$^{-1}$ collagenase P and digested for 20 min at 37 °C. Following digestion, the pancreas was triturated and centrifuged for 1 min at $200 \times g$. The pellet was resuspended in 10 ml HBSS 10% FBS, triturated and centrifuged again. This washing/dispersion step was repeated for four times and the remaining pellet was resuspended in 1 ml TRIzol. Total RNA was extracted using Phase Lock Gel (VWR) following the manufacturer's recommendations. Two micrograms of total RNA were then digested with DNAse I and reverse-transcribed with Superscript II using oligoDT primer (ThermoFisher Scientific) according to the supplier's instructions. Quantitative PCR was performed with the following primers: OVA mRNA forward 5′-CTTGAGCAGCTTGAGAGTATAA-3′, OVA mRNA reverse 5′-CCATCTTCATGCGAGGTAAG-3′, HPRT mRNA forward 5′-CTCCTCGAGAC CGCTTTTTGC-3′, HPRT mRNA reverse 5′-TAACCTGGTTCATCATCGCTAA TC-3′, CRP mRNA forward 5′-TCAGCTTCTCTCGGACTT-3′, CRP mRNA reverse 5′-CTGCTTCCAGAGACACATAG-3′, INS2 mRNA forward 5′-GTGGC TTCTTCTACACACC-3′, INS2 mRNA reverse 5′- TACAATGCCACGCTTCT G-3′. Each reaction was performed in triplicate using a real-time cycler ABI Prism 7500 (Life Technologies) and Power SYBR Green (ThermoFisher Scientific). No RT controls were performed in parallel to confirm the absence of genomic DNA contamination. Results are reported as relative expression as compared to the WT LMC negative control (negative values were set to a Ct value of 40) and were normalized to HPRT.

**Tetramer-based enrichment of antigen-specific T cells**. The spleen and lymph nodes of six to twelve week-old naïve mice were harvested and dissociated using frosted glass slides, and the single-cell suspension was passed through nylon mesh. The cells were washed with cold sort buffer (PBS 2% FBS), and the cell pellet was resuspended in Fc block. The cells were then stained with PE- and APC-labeled tetramers. Finally, the tetramer positive cells were enriched using Miltenyi anti-PE microbeads and LS magnetic columns (Miltenyi Biotec).

**In vitro culture of sorted mTEC with thymocytes**. mTEC$^{lo}$ (EpCAM$^+$CD45$^-$UEA-1$^+$Ly51$^-$CD80$^{lo}$MHC-II$^{lo}$) and mTEC$^{hi}$ (EpCAM$^+$CD45$^-$UEA-1$^+$Ly51$^-$CD80$^{hi}$MHC-II$^{hi}$) cells were sorted from five to ten week-old male or female WT

and CRP Tg mice. mTEC, WT thymocytes, and OT-I thymocytes (mixed 1:1:1 for a total of $10^5$-$10^6$ cells) were combined, and the cells were pelleted. The supernatant was carefully removed, and the pellet was vortexed to form a slurry which was deposited in a standing drop on a cell culture insert over RPMI 10% FBS. The cells were incubated at 37 °C, 5% $CO_2$ for 24 h.

**Single-cell RNA sequencing**. Thymic epithelial cells (CD45$^-$EpCAM$^+$) were sorted from 9 week-old female CRP, CRP$^{lo}$ and INS2 Tg mice, and a 10X Genomics Single-Cell 3′ transcriptome library was prepared (Génome Québec). The libraries were sequenced using the Illumina HiSeq4000 PE100. Cell Ranger version 3.0.1 (10X Genomics) was used with default parameters to demultiplex sequencing reads, align them to a custom reference genome, distinguish cells from background, and obtain gene counts per cell. Reads were aligned to the mm10 reference genome provided by 10X modified to include the sequence of the transgene (Fig. 1a). The Ensembl transcriptome (v93) was used for counting, and, for each sample, cells were filtered on the following quality control (QC) metrics: mitochondrial content (indicative of cell damage), number of genes, and number of unique molecular identifiers (UMIs), using the Seurat package (v2.3)[74]. Thresholds for each sample are provided in Supplementary Table 1 and were selected according to the distribution of each metric within the sample, which varies with sequencing coverage and the number of cells captured. QC metrics statistics before and after filtering are provided in Supplementary Table 1, as well as in Supplementary Fig. 1g.

Samples were then processed using Seurat and cytobox[75]. Libraries were scaled to 10,000 UMIs per cell and log-normalized. UMI counts and mitochondrial content were regressed from normalized gene counts and the residuals z-scored gene-wise. Dimensionality reduction was performed using principal component analysis (PCA) applied to the most variant genes, and PCA was used as input for projection to two dimensions using uniform manifold approximation and projection (UMAP) and clustering. Clusters were identified using the shared nearest neighbor (SNN) modularity optimization-based clustering algorithm provided by Seurat, on the first 30 principal components, with a resolution of 0.6.

**Tumor cell inoculation**. $5 \times 10^5$ B16 or B16-OVA melanoma cells were injected subcutaneously in the right flank of six to twelve week-old male or female mice, and tumor area was measured every 2 days starting 7 days post-inoculation. Tumor growth was followed by measuring tumor diameter using a caliper and calculation of the tumor area. Mice were euthanized by cervical dislocation if tumors reached a diameter of 17 mm or in case of ulceration as requested by the institutional ethical board. For some experiments, the mice were injected with 200 μg α-CD25 antibody at day −4 and −1 in order to deplete CD25$^+$ T$_{reg}$ prior to tumor inoculation.

**Statistical analysis**. Each experiment was repeated independently with similar results at least two times except for single cell RNA sequencing. For comparison between two groups, data with a normal distribution were analyzed for statistical significance using the unpaired Student $t$-test while data without a normal distribution were analyzed with the two-tailed Mann–Whitney U test. Data from more than two groups were analyzed with Kruskal–Wallis test followed by Dunn's post-hoc comparisons to LMC, two-sided. Tumor growth was subjected to a linear mixed effect modeling applied to log pre-processed tumor surfaces. All values where $p < 0.05$ were considered statistically significant. Analyses were performed with GraphPad Prism version 6. $*p < 0.05$, $**p < 0.01$, $***p < 0.001$.

**Reporting summary**. Further information on research design is available in the Nature Research Reporting Summary linked to this article.

## Data availability

The single-cell RNA-sequencing data have been deposited in NCBI Gene Expression Omnibus (GEO) and is accessible through GEO Series accession number GSE153288 [https://www.ncbi.nlm.nih.gov/geo/query/acc.cgi?acc=GSE153288]. The source data underlying Fig. 1b–d, g–i, k–m, Fig. 2b–f, Fig. 3b, d, f, g, i–l, Fig. 4a, b, d, e, Fig. 5b, c, e, g, i, Fig. 6b, d, f, h–k, Fig. 7a–e, Supplementary Fig. 1c, d, e, Supplementary Fig. 2 d–g, i–p, Supplementary Fig. 3c, e, f, Supplementary Fig. 4e, f, and Supplementary Fig. 5e are provided as a Source Data file. All other data supporting the findings of this study are available within the article and its supplementary information files, or are available upon request to the authors. Source data are provided with this paper.

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

## Acknowledgements

We acknowledge Michael Zeidler and Thomas Saunders for BAC Recombineering and the Transgenic Animal Model Core of the University of Michigan's Biomedical Research Core Facilities. We thank Dr. Benjamin Turgeon and Dr. Marilaine Fournier for assistance in generating the BAC transgenics as well as Dr. Nathalie Labrecque and Jean-François Daudelin as well as members of the Lesage and Melichar labs for helpful discussions and reagents. We are grateful to Drs. Ellen Robey, Sylvie Lesage, Troy Baldwin, Nathalie Labrecque, Lauren Ehrlich, and Marilaine Fournier as well as Ms. Mengqi Dong for critically reading the manuscript. We thank Martine Dupuis for assistance with flow cytometry and cell sorting as well as the staff of the animal facilities for the maintenance of the mouse colonies at Maisonneuve-Rosemont Hospital Research Center. This work was supported by a grant from the Canadian Institutes of Health Research (MOP-142254) awarded to H.J.M. M-È.L. is supported by FRQS and L'Oréal-UNESCO For Women in Science post-doctoral fellowships. M.G. received a Diabète Québec summer fellowship. H.J.M. is a junior 1 scholar of the FRQS, a CIHR New Investigator (MSH-141967), and a Cole Foundation Early Career Transition award recipient.

## Author contributions

M.-È.L. and H.J.M. conceived and designed experiments. M.È.L. and M.G. performed the experiments. M.C. and C.L.K. directed and analyzed the single cell RNA sequencing experiment. J.J.M. designed the universal self-antigen and provided reagents. M.-È.L. and H.J.M. wrote the paper. All authors revised the manuscript.

## Competing interests

The authors declare no competing interests.
