## [Peer Review File · Nature Communications]

Reviewers' comments:

Reviewer #1 (Thymic selection, TEC) (Remarks to the Author):

The MS addresses a highly interesting question: does self-antigen expression in distinct mTEC subsets result in different modes of central tolerance. The authors address this question through expression of a 'multiepitopic' model antigen from transgenes that either contain the CRP or Ins2 regulatory elements, resulting in expression in both mTEC_{lo} and mTEC_{hi} cells or selective expression only in mTEC_{hi} cells, respectively. The data that are presented are consistent with a scenario whereby antigen expression in mTEC_{lo} (but not mTEC_{hi}) induces diversion into the CD8_{aa} IEL lineage among MHC_I restricted thymocytes, whereas for MHC_{II} restricted T cells, expression of cognate antigen solely by mTEC_{hi} induces specific Foxp3⁺ Treg cells. This is an intriguing suggestion that should be of great interest to the community. However, a major concern is that a comparison of transgene expression in bulk sorted cells may not reveal differences/variations of antigen expression among individual cells within the respective cell population. It is known that typical Aire-dependent TRAs (such as Ins2) are expressed by only a small percentage of mTEC_{hi} cells, while this is less clear for Aire independent TRAs. Curiously, the CRP transgene is expressed at 'identical' levels in bulk mTEC_{hi} as compared to Ins2 mTEC_{hi}, yet only the later generates specific Treg cells. The authors do discuss this, but the suggested explanations remain very speculative. Could it be that the lack of Treg induction in the CRP model is a consequence of very little antigen expression in many mTEC_{hi} cells (as opposed to relatively high antigen expression in only a few mTEC_{hi} cells in the Ins2 model)? Have the authors tried to address the expression of the model antigen at the single cell level by analyzing the 'linked' GFP fluorescence or single cell PCR?

Other points:

Fig 1a: It is difficult to understand how the model antigen is ultimately expressed in the mTECs. As a membrane embedded molecule with the antigenic regions facing the extracellular space?

Fig 3: Isn't the presence of a substantial population of IEL_p cells (CD112+PD-1+TCR_b+DN) among OT-1⁺ cells in the absence of cognate antigen (i.e. in LMCs) surprising? Is this a peculiar property of this TCR transgene or has this been observed for other MHC_I restricted TCR transgenes as well? These cells (at least as far as the PD-1⁺ IEL_p subset is concerned) are believed to be agonist induced, so why do they exist in LMCs? This point is of importance because it raises the question whether the apparently 'induced' IEL_p cells in CRP mice are actually increased in numbers or may simply be protected from negative selection and only seemingly increased (in % among OT-1 cells). Would be helpful to include absolute cell numbers here.

Fig 5d: Is it established that MHC_I tetramers efficiently stain cells lacking CD8_b or even all CD8 molecules (e.g. would OT1 IELs that are either CD8_{ab} or CD8_{aa} or CD8-negative be equally stained with a tetramer?)?

Fig 5f: Do these tetramers (APC and PE) both work? There is a distinct population of TetPE⁺/TetAPC⁻ cells that looks more convincing as the dispersed double-positive 'population'?

Fig 6g: The CD25/Foxp3 staining is not really convincing.

Reviewer #2 (Aire, TEC, repertoire) (Remarks to the Author):

The manuscript from Lebel et al is an interesting study that attempts important experiments to unravel the role of the thymus medulla in tolerance induction. To do this, the authors generate new transgenic mice in which a model antigen (OVA) is expressed under the control of the promoter of an Aire-dependent TRA (Ins2) or an Aire-independent TRA (CRP). The idea here is that the differing patterns of expression of the self-antigen by different mTEC subsets can then be assessed using OVA-specific TCR transgenic thymocytes (and also OVA-specific tetramers), to see how antigen presentation in the medulla impacts negative selection, and lineage divergence eg Treg generation, CD4-8-TCR+ and CD8aa T-cell generation. Overall, the study is of interest as it tackles poorly understood questions, and it also generates new models that have the potential to advance the field. Despite this, there are a number of points that require clarification and improvement.

1. In figure 1, the transgenic constructs include eGFP, presumably to be able to identify mTEC that express OVA in the two different strains. Can GFP be detected and if so, the data should be included.
2. In many of the experiments to study thymic tolerance, the authors make use of in vitro slices of thymus cultures, and bone marrow transplants. While the bone marrow transplant experiment has the advantage of studying selection in vivo, point 3 needs to be clarified. Also, it is unclear how the thymic slice system relates to the thymus in vivo. Many of the concerns from these experiments would be simplified if the authors crossed their antigen transgenic mice to OT1. Has this been done?
3. In the bone marrow chimeras, is a congenic system of CD45.1/CD45.2 host/donor used? This is essential to really discriminate between donor T-cell development and radio resistant host cells.
4. The experiments to look at negative selection, Treg generation, CD8aa T-cell generation show only % of cells, inclusion of absolute numbers of cell types would be needed to strengthen the case.

Reviewer #3 (Thymic selection, Aire, autoimmunity) (Remarks to the Author):

This is a study from Lebel et al that is focused on examining in more detail thymic selection by thymic epithelial cells using novel BAC transgenic mice driving expression a panel of neo-antigens (like ovalbumin/OVA). The group used two different BAC's to drive expression: CRP and the Ins2 gene and they then drive the expression of a novel gene cassette that contains OVA along with several other epitomes that can be measured by tetramer analyses. They argue that CRP driven and Ins2 driven expression is distinct in TEC's with CRP in both mTEC_{lo} and mTEC_{hi} and Ins2 only in mTEC_{hi}. From there, they explore deletion of thymocytes with OT-1 CD8 T cells using a thymic slice approach and find that both BAC trans genes can drive deletion and induction of CD8aa cells with some subtle differences between the two. Next they explore OT-II CD4 T cell selection by using a mixed bone marrow chimera approach of WT plus OT-II bone marrow cells put into irradiated BAC Tg hosts. Here, they argue that Ins2 driven expression can promote Treg induction over the CRP transgene but somewhat paradoxically they don't observe deletion of OT-II cells. Next they move to the polyclonal repertoire using sensitive tetramers and find evidence of deletion of CD8's and induction of Treg's in Ins2 mice. Finally, they then move to functional consequences of this selection phenomenon by immunizing mice or giving them tumors and get evidence of some tolerance when trans genes are present.

This is a complex and interesting paper that revisits the topic of thymic negative selection and if selected cell types in the medulla can promote deletion vs Treg induction. There is a large body of literature in this area using similar approaches with trans genes, knockins, and BAC transgenes. For the most part, this particular study generally ignores this literature and the current models on this topic and just calls the results "inconsistent". Thus, it is difficult to tease out what is really new here or how it compares to this Literature. Indeed, this is the major weakness of the paper. The broad conclusions that can be reached here are difficult to really tease out. For example, It's previously known that if you target a neo-antigen to mTEChi via Aire, you can promote Treg selection such as to HA (Klein et al. Nat Immunology). Likewise, Treg induction also appears to rely on relative properties of the TCR sequence and the APC's as shown by the Hsieh group at Wash U and the Savage group at the U of Chicago. How do the results here, really provide any new broad clarity to the previous work?

The latter part of the paper with giving mice tumors and infections give predictable results and really don't add much to what the paper is trying to say as it relates to thymic selection and tolerance.

There are also many technical complexities to this work which at this point, limit some of the major interpretations:

- 1) A more thorough work up of the pattern of expression of the BAC transgenes needs to be presented across the entire animal. Are CRP BAC Tg expressed in liver? Which cells? Likewise Ins2 BAC Tg expression in pancreatic islets/ beta cells? Does this peripheral expression influence the results here? How does thymic expression compare in these tissues to thymic expression? What about expression in cortical TEC's and DC's and other APC's in the thymus? Furthermore, the peripheral expression could play a major role in the polyclonal tetramer experiments here. For instance the Treg's could be iTreg's vs thymic ally derived Treg's in the pull downs in Figure 5.
- 2) Why the thymic slice approach for negative selection with OT-1 and mixed chimeras for OT-II? Why not just genetically cross the TCR Tg to the BAC Tg? The use of irradiation and/or looking at slices could have their own technical limitations.
- 3) In Figure 4, two major issues: a) Why no negative selection? Need to see flow data for CD4/CD8 stain in thymus and OT-II clonotype. B) Using % of CD4's is not enough, need to show absolute numbers of Treg's. Previous work, such as in Ref. #22 has shown that when deletion and Treg induction are mixed together absolute numbers of Treg's are unchanged in OT-II x RipMova mice.
- 4) For CRP Tg mice: Which TEC cells are doing the selection? How can you really tell?
- 5) In polyclonal mice can thymi be pooled from multiple mice to look at tetramer positive cells??

Minor comment:

Figure 5: What are the PE positive cells that are APC negative in panel f?

Figure 5: Cell numbers in panel H are very low (as expected), how many mice were used? Is this really statistically significant? What about absolute numbers of cells (this is very important)? Finally, are these iTreg's vs thymic Treg's???

Reviewer #1 (Thymic selection, TEC) (Remarks to the Author):

The MS addresses a highly interesting question: does self-antigen expression in distinct mTEC subsets result in different modes of central tolerance. The authors address this question through expression of a 'multiepitopic' model antigen from transgenes that either contain the CRP or *Ins2* regulatory elements, resulting in expression in both mTEC^{lo} and mTEC^{hi} cells or selective expression only in mTEC^{hi} cells, respectively. The data that are presented are consistent with a scenario whereby antigen expression in mTEC^{lo} (but not mTEC^{hi}) induces diversion into the CD8aa IEL lineage among MHCII restricted thymocytes, whereas for MHCI restricted T cells, expression of cognate antigen solely by mTEC^{hi} induces specific Foxp3⁺ Treg cells. This is an intriguing suggestion that should be of great interest to the community. However, a major concern is that a comparison of transgene expression in bulk sorted cells may not reveal differences/variations of antigen expression among individual cells within the respective cell population. It is known that typical Aire-dependent TRAs (such as *Ins2*) are expressed by only a small percentage of mTEC^{hi} cells, while this is less clear for Aire independent TRAs. Curiously, the CRP transgene is expressed at 'identical' levels in bulk mTEC^{hi} as compared to *Ins2* mTEC^{hi}, yet only the later generates specific Treg cells. The authors do discuss this, but the suggested explanations remain very speculative. Could it be that the lack of Treg induction in the CRP model is a consequence of very little antigen expression in many mTEC^{hi} cells (as opposed to relatively high antigen expression in only a few mTEC^{hi} cells in the *Ins2* model)?

1. Have the authors tried to address the expression of the model antigen at the single cell level by analyzing the 'linked' GFP fluorescence or single cell PCR?

We thank the reviewer for this suggestion. Indeed, we attempted to assess transgene expression (GFP and anti-OVA Ab) at the single cell level by flow cytometry, but this technique was not sufficiently sensitive to draw any conclusions.

Instead, we performed genome-wide, single-cell RNA sequencing on total sorted thymic epithelial cells (TEC) from CRP, CRP^{lo} and *INS2* Tg mice. This analysis is now included in the manuscript (Fig. 1 and Supplementary Fig. 1). We were able to detect transgene expression in TEC from CRP mice. However, transgene expression was not detected in TEC from CRP^{lo} mice and was very low in TEC isolated from *INS2* mice. The single-cell sequencing data allowed us to analyze endogenous TRA expression patterns across cells to address the important point raised by the reviewer. Indeed, expression analysis at the single-cell level uncovered differences that were missed in the bulk transcriptomic data. *Crp* is expressed in a greater proportion of mTEC^{hi}, but at relatively lower levels on a per cell basis, while endogenous *Ins2* is expressed in very few mTEC^{hi} cells, but at a higher level in individual cells (Fig. 1h-i). Thus, it is possible that T_{reg} differentiation requires higher antigen expression on a per cell basis. Additional discussion to this effect is now included in the manuscript.

In addition, the higher endogenous *Crp* and CRP transgene (USA) expression in bulk mTEC^{lo} vs mTEC^{hi}, determined by qPCR (Supplementary Fig. 1b and Fig. 1b, c), is further explained by the

single-cell RNA sequencing data; a higher proportion of mTEC^{lo} express *Crp* and the CRP transgene, but *Crp* and CRP transgene expression in individual cells is similar between these two mTEC subtypes (Fig. 1h-i).

Of note, we refrained from reporting comparisons between the transgenes, since the coverage of the transgene in INS2 Tg mice was too low to report. This is not surprising since single-cell sequencing data has a high-dropout rate and transgene expression was expected, by design, to be at endogenous levels or lower (which was the case). We hope the reviewer will agree with our decision to be stringent and avoid a formal comparison between transgenes.

2. Fig 1a: It is difficult to understand how the model antigen is ultimately expressed in the mTECs. As a membrane embedded molecule with the antigenic regions facing the extracellular space?

Yes, the construct is designed such that OVA and the additional epitopes are in the extracellular domain¹. Dr. Moon's group has been able to demonstrate this via surface staining of highly expressing transgenic cells with an OVA antibody (unpublished data). The orientation of the transgene is now clarified in the manuscript.

3. Fig 3: Isn't the presence of a substantial population of IELp cells (CD112+PD-1+TCRb+DN) among OT-1+ cells in the absence of cognate antigen (i.e. in LMCs) surprising? Is this a peculiar property of this TCR transgene or has this been observed for other MHC I restricted TCR transgenes as well? These cells (at least as far as the PD-1+ IELp subset is concerned) are believed to be agonist induced, so why do they exist in LMCs? This point is of importance because it raises the question whether the apparently 'induced' IELp cells in CRP mice are actually increased in numbers or may simply be protected from negative selection and only seemingly increased (in % among OT-1 cells). Would be helpful to include absolute cell numbers here.

The proportion of CD4⁺CD8⁻ DN cells among the T cell population in OT-I mice is quite low (~1-2%). Thus, the higher proportion of CD4⁺CD8⁻ DN cells observed in LMC bone marrow chimera (BMC) might be a consequence of this model. To that end, we observe a similar proportion of CD4⁺CD8⁻ DN cells when another TCR transgene (P14) is used to generate low percentage hematopoietic chimeras in wild-type mice (10% DN with P14 [Supplementary Fig. 2f] vs 20% DN with OT-I [Fig. 3f]).

Importantly, in the initial manuscript, we presented the total proportion of TCR^β^{hi}CD4⁺CD8⁻ DN cells, not the PD1⁺ population, for the BMC data. The absolute cell numbers of CD122⁺PD-1⁺TCR^β^{hi}CD4⁺CD8⁻ DN cells, in the context of the low percentage hematopoietic chimera, is now added to Fig. 3 and clearly shows a significant increase in the PD1⁺ population in the CRP BMC as compared to controls. For the thymic slices, the cell number calculation is not appropriate as the size of the slices can be quite variable from sample to sample; thus, this data is not included in the revised manuscript.

4. Fig 5d: Is it established that MHC I tetramers efficiently stain cells lacking CD8b or even all CD8 molecules (e.g. would OT1 IELs that are either CD8ab or CD8aa or CD8-negative be equally stained with a tetramer?)?

Conflicting data have been reported regarding the requirement of CD8 $\alpha\beta$ for efficient tetramer binding. For example, it has been shown that CD8 down-regulation during acute infection correlates with decreased pMHC multimer binding². In addition, staining with some CD8 antibodies hinders

subsequent tetramer staining³. On the other hand, CD8 affinity for pMHC-I is relatively weak, and it has been shown that CD8 $\alpha\alpha$ and CD8 $\alpha\beta$ coreceptors bind to pMHC-I with similar affinity^{4,5}. Moreover, some tetramers were shown to bind as efficiently to CD8 β KO T cells as to their wild-type counterparts, and other tetramers that do not interact with CD8 efficiently identify antigen specific CD8 T cells⁶.

If we stain OT-I Tg IELs from chimeric mice with the H2K^b-OVA tetramer, it is possible to see that both CD8 $\alpha\alpha$ and CD8 $\alpha\beta$ T cells are stained by the tetramer (see the figure below). However, if we compare the mean fluorescence intensity (MFI) for the tetramer staining, it is lower in the CD8 $\alpha\alpha$ population suggesting that the CD8 $\alpha\beta$ coreceptor is not necessary, but may increase tetramer binding. Nevertheless, in our experiments, we compare the number of tetramer⁺ cells rather than the MFI of the tetramer staining.

As such, in our experiments, we cannot rule out that tetramer binding is less efficient on cells lacking CD8 β or all CD8 molecules as it is on conventional antigen-specific CD8⁺ T cells. However, consistent with our TCR Tg data (OT-I), we observed an increase in the relative number of polyclonal OVA specific T cells (not a decrease) within the intraepithelial lymphocyte population of CRP mice of which an increased proportion are CD8 $\alpha\alpha$ suggesting that the tetramer is able to bind these cells (Fig. 5c-e).

Response to reviewer Figure 1. H2K^b-OVA tetramer staining in OT-I Tg⁺ IELs isolated from a wild type bone marrow chimera and gated on the CD8 $\alpha\beta$ or CD8 $\alpha\alpha$ population from the polyclonal (CD45.1.2, black) and OT-I (CD45.1, gray) cells. MFI: mean fluorescence intensity.

5. Fig 5f: Do these tetramers (APC and PE) both work? There is a distinct population of TetPE+/TetAPC- cells that looks more convincing as the dispersed double-positive 'population'?

Both PE- and APC-conjugated tetramers were tested/titrated beforehand by staining SMARTA Tg T cells diluted in wild-type polyclonal splenocytes to confirm their efficacy. The population of tetPE+/tetAPC- cells in the original panel was an artifact of the PE enrichment that was performed. As such, only the double positive cells are considered true antigen-specific cells.

More specifically, a few of our experiments included a PeCy7-labeled antibody in the mix, and it is possible that the PeCy7 antibody binds to the PE beads during the PE enrichment leading to the appearance of a PE positive population due to some fluorochrome uncoupling. This PE^{hi} single positive population only occurs in the presence of the PeCy7 antibody. We have now replaced these plots with representative profiles from an I-A^b-gp66 tetramer staining experiment in which no PeCy7-conjugated antibody was included. The inclusion/exclusion of the PeCy7-conjugated antibody did not influence the results of this experiment.

6. Fig 6g: The CD25/Foxp3 staining is not really convincing.

We agree that the original flow cytometry profiles that we included in the manuscript were not optimal. Although they were representative of our results in terms of the proportion of T_{reg} cells

generated in these models, the figure has been updated with more convincing representative flow plots.

Reviewer #2 (Aire, TEC, repertoire) (Remarks to the Author):

The manuscript from Lebel et al is an interesting study that attempts important experiments to unravel the role of the thymus medulla in tolerance induction. To do this, the authors generate new transgenic mice in which a model antigen (OVA) is expressed under the control of the promoter of an Aire-dependent TRA (Ins2) or an Aire-independent TRA (CRP). The idea here is that the differing patterns of expression of the self-antigen by different mTEC subsets can then be assessed using OVA-specific TCR transgenic thymocytes (and also OVA-specific tetramers), to see how antigen presentation in the medulla impacts negative selection, and lineage divergence eg Treg generation, CD4-8-TCR+ and CD8aa T-cell generation. Overall, the study is of interest as it tackles poorly understood questions, and it also generates new models that have the potential to advance the field. Despite this, there are a number of points that require clarification and improvement.

1. In figure 1, the transgenic constructs include eGFP, presumably to be able to identify mTEC that express OVA in the two different strains. Can GFP be detected and if so, the data should be included.

Unfortunately, we are not able to detect GFP⁺ cells by flow cytometry. We have included additional characterization of the transgenes and their endogenous gene counterparts using qPCR and single-cell RNA sequencing (Fig. 1 and Supplementary Fig. 1).

2. In many of the experiments to study thymic tolerance, the authors make use of in vitro slices of thymus cultures, and bone marrow transplants. While the bone marrow transplant experiment has the advantage of studying selection in vivo, point 3 needs to be clarified. Also, it is unclear how the thymic slice system relates to the thymus in vivo. Many of the concerns from these experiments would be simplified if the authors crossed their antigen transgenic mice to OT1. Has this been done?

Our rationale for the thymic slice and low percentage BMC approaches used in the original manuscript was that it has been shown that a too high number of TCR Tg cells can bias the selection process because of competition between thymocytes for their cognate ligand and cytokines. For example, this intraclonal competition affects T_{reg} differentiation⁷⁻⁹.

Thymic slice organotypic cultures are a versatile model to study thymic selection *in situ*¹⁰. This model has been shown to preserve the integrity of the thymic cortical and medullary regions thus providing a framework of stromal cells that supports thymocyte migration as well as efficient positive and negative selection, T_{reg} differentiation⁷, and pIEL generation¹¹.

However, we agree that these two models are not perfect particularly in terms of radiation sensitive APCs in the low percentage BMC model. Therefore, we crossed our TRA Tg mice with OT-I as the reviewer requested, and we obtained the same phenotypes as reported in the original manuscript with the two models described above. These results are now included in Supplementary Fig. 2.

3. In the bone marrow chimeras, is a congenic system of CD45.1/CD45.2 host/donor used? This is essential to really discriminate between donor T-cell development and radio resistant host cells.

Yes, a congenic system of CD45.1/CD45.2 or Thy1.2/Thy1.1 host/donor was used in the BMC experiments. This point has been clarified in the figure legend. In addition, the gating strategy has now been added to Supplementary Fig. 2 and 4.

4. The experiments to look at negative selection, Treg generation, CD8aa T-cell generation show only % of cells, inclusion of absolute numbers of cell types would be needed to strengthen the case.

We agree with the reviewer's comment. As such, absolute numbers of cell populations were added for most experiments. We did not include absolute numbers for the thymic slice experiments as this can vary significantly depending on the size of individual slices.

Reviewer #3 (Thymic selection, Aire, autoimmunity) (Remarks to the Author):

This is a study from Lebel et al that is focused on examining in more detail thymic selection by thymic epithelial cells using novel BAC transgenic mice driving expression a panel of neo-antigens (like ovalbumin/OVA). The group used two different BAC's to drive expression: CRP and the Ins2 gene and they then drive the expression of a novel gene cassette that contains OVA along with several other epitomes that can be measured by tetramer analyses. They argue that CRP driven and Ins2 driven expression is distinct in TEC's with CRP in both mTEC_{lo} and mTEC_{hi} and Ins2 only in mTEC_{hi}. From there, they explore deletion of thymocytes with OT-1 CD8 T cells using a thymic slice approach and find that both BAC trans genes can drive deletion and induction of CD8aa cellswith some subtle differences between the two. Next they explore OT-II CD4 T cell selection by using a mixed bone marrow chimera approach of WT plus OT-II bone marrow cells put into irradiated BAC Tg hosts. Here, they argue that Ins2 driven expression can promote Treg induction over the CRP transgene but somewhat paradoxically they don't observe deletion of OT-II cells. Next they move to the polyclonal repertoire using sensitive tetramers and find evidence of deletion of CD8's and induction of Treg's in Ins2 mice. Finally, they then move to functional consequences of this selection phenomenon by immunizing mice or giving them tumors and get evidence of some tolerance when transgenes are present.

This is a complex and interesting paper that revisits the topic of thymic negative selection and if selected cell types in the medulla can promote deletion vs Treg induction. There is a large body of literature in this area using similar approaches with trans genes, knockins, and BAC transgenes. For the most part, this particular study generally ignores this literature and the current models on this topic and just calls the results "inconsistent". Thus, it is difficult to tease out what is really new here or how it compares to this Literature. Indeed, this is the major weakness of the paper. The broad conclusions that can be reached here are difficult to really tease out. For example, It's previously known that if you target a neo-antigen to mTEC_{hi} via Aire, you can promote Treg selection such as to HA (Klein et al. Nat Immunology). Likewise, Treg induction also appears to rely on relative properties of the TCR sequence and the APC's as shown by the Hsieh group at Wash U and the Savage group at the U of Chicago. How do the results here, really provide any new broad clarity to the previous work?

Our intention was not to suggest that the body of literature concerning negative selection versus T_{reg} induction was inconsistent. The 'inconsistency' we referred was related to the fact that minimal promoters used for transgenes can be influenced by a number of factors (copy number, insertion site, etc.) and that the same minimal promoter in different transgenic lines can lead to different patterns of expression (some of the examples given were specifically for the insulin promoter). This was used as rationale as to why we chose a BAC Tg approach for model antigen expression. We have adjusted our wording so that this is not misconstrued going forward.

In addition, our intention is not to ignore previous studies but to address additional factors that may influence autoreactive T cell fate in the thymus. Indeed, we add additional dimensions to the discussion here in terms of the role of different mTEC subsets, and, perhaps, the level of expression of a given tissue restricted antigen on a per cell basis in influencing negative selection versus IELp or T_{reg} generation. We have added context for our results by including additional, relevant references to the literature in the discussion section.

The latter part of the paper with giving mice tumors and infections give predictable results and really don't add much to what the paper is trying to say as it relates to thymic selection and tolerance.

We agree that the main goal of the paper was to better understand thymic selection. However, we think that showing the biological consequences of these differences in thymic selection further solidifies the results and adds a functional perspective. In addition, although it is predictable that the presence of OVA specific T_{reg} interfere with B16-OVA tumor control, we think that it is interesting to demonstrate that the AIRE-independent TRA expression in this model does not induce tolerance mechanisms that affect tumor control. This could be relevant to re-activating T cells specific for self-peptides. Finally, given that the function of TCR $\alpha\beta$ CD8 $\alpha\alpha$ IELs is still a matter of debate, we consider the results of the intestinal infection contribute to our understanding of these unconventional T cells.

There are also many technical complexities to this work which at this point, limit some of the major interpretations:

1) A more thorough work up of the pattern of expression of the BAC transgenes needs to be presented across the entire animal. Are CRP BAC Tg expressed in liver? Which cells? Likewise Ins2 BAC Tg expression in pancreatic islets/ beta cells? Does this peripheral expression influence the results here? How does thymic expression compare in these tissues to thymic expression? What about expression in cortical TEC's and DC's and other APC's in the thymus? Furthermore, the peripheral expression could play a major role in the polyclonal tetramer experiments here. For instance the Treg's could be iTreg's vs thymic ally derived Treg's in the pull downs in Figure 5.

We added additional data regarding transgene expression by APC isolated from the thymus as well as peripheral expression in the liver, pancreas, and spleen. As expected, OVA is expressed in the liver of the CRP mice while it is expressed in the pancreas of the INS2 mice, and no expression was detected in the spleen in either of the Tg backgrounds. In addition, no transgene expression is detected in B cells, DC, or macrophages isolated from the thymus of the CRP or INS2 Tg mice. These data are now included in Supplementary Fig. 1.

Moreover, we performed single-cell RNA sequencing analysis on total sorted thymic epithelial cells (TEC) from CRP, CRP⁰ and INS2 Tg mice. We were able to detect transgene expression in TEC from CRP mice. However, transgene expression was not detected in TEC from CRP⁰ mice and very low in TEC isolated from INS2 Tg mice. While the transgene expression signal was not sufficient to perform a reliable comparison between the three transgenic mice, single-cell data allowed us to analyze endogenous TRA expression patterns across cells to address the important point raised by the reviewer. Indeed, expression analysis at the single-cell level uncovered differences that were missed in the bulk transcriptomic data. The results of our analysis of the transgenes in CRP Tg mice as well as their endogenous gene counterparts in TEC isolated from the three Tg mice is now also included in the manuscript (Fig. 1 and Supplementary Fig. 1). According to this analysis, model antigen expression in cTEC is minor as compared to the mTEC populations (Fig. 1g).

We now also include data regarding the impact of peripheral expression of the transgene on T_{reg} differentiation in Fig. 6. More specifically, Aire^{-/-} INS2 mice, in which the transgene expression is lost in the thymus, but is still present in the periphery, were infected with LCMV-OVA, and the virus-specific immune response was analyzed. These data suggest that expression of model antigen in peripheral tissues (e.g. pancreas) is sufficient to limit the OVA-specific effector CD8⁺ T cell response. However, loss of transgene expression in the thymus completely abrogates the impact of model antigen expression on the gp66-specific CD4⁺ T cell response and the proportion of T_{reg} suggesting that transgene expression in the thymus is required to induce T_{reg} differentiation.

In addition, we performed adoptive transfer of SMARTA or OT-II TCR Tg cells in control or INS2 mice and performed an LCMV infection two weeks later. In this context, we do not observe any increase in T_{reg} numbers in INS2 mice suggesting that peripheral expression of the model antigen is not sufficient to induce T_{reg} differentiation. We provide this data for the reviewer in the figure below, but it is not included in the revised manuscript as these experiments have only been performed once for each transgene and are consistent with the strong data from the Aire-deficient TRA Tg mice.

Response to reviewer Figure 2. Adoptive transfer of SMARTA or OT-II Tg CD4⁺ T cells into INS2 Tg mice does not lead to an increase in the proportion of T_{reg}. (a) Representative flow plots of CD25 and FoxP3 staining and (b) compilation of the proportion of CD25⁺FoxP3⁺ SMARTA Tg CD4⁺ T cells. (c) Representative flow plots of CD25 and FoxP3 staining and (d) compilation of the proportion of CD25⁺FoxP3⁺ OT-II Tg CD4⁺ T cells.

2) Why the thymic slice approach for negative selection with OT-1 and mixed chimeras for OT-II? Why not just genetically cross the TCR Tg to the BAC Tg? The use of irradiation and/or looking at slices could have their own technical limitations.

Our rationale for the thymic slice and low percentage BMC approaches used in the original manuscript was that it has been shown that a too high a number of TCR Tg cells can bias the selection process because of competition between thymocytes for their cognate ligand and cytokines. For example, this intraclonal competition affects T_{reg} differentiation⁷⁻⁹. Thus, in order to avoid this undesirable effect, we decided to use the thymic slice system and low-density BMC models. In addition, we validated many of the results using tetramer staining of endogenous, polyclonal T cells.

However, we agree that these two models have some caveats. Thus, we crossed our Tg mice with OT-I TCR Tg mice and the phenotypes were consistent with those presented in the original manuscript. These results have been added to Supplementary Fig. 2 of the revised manuscript.

However, in the case of the OT-II TCR Tg mice, crossing them to the BAC Tg mice did not result in an increase in T_{reg} in the INS2 Tg strains. This may be due, as previously published by others, to an inverse correlation between the number of TCR Tg cells and their capacity to differentiate in T_{reg} .

Alternatively, chimeras were generated in which OT-II Tg bone marrow was injected into non-irradiated CRP and INS2 Tg neonates. With this model, we are able to generate chimeras with a low proportion of OT-II cells in the absence of any damage to radiation-sensitive APCs. As you can see below, a relatively higher proportion of OT-II Tg T_{reg} are detected in INS2 Tg mice but not in CRP Tg mice, thus confirming the results we obtained using the low density hematopoietic chimeras and the tetramer staining of endogenous polyclonal cells. Notably, as the engraftment is variable from mouse to mouse, we cannot confidently assess OT-II T cell deletion in this model.

Response to reviewer Figure 3. INS2 and CRP neonates (Thy1.2, 2-4 days of age) were injected with 5×10^6 OT-II (Thy1.1) bone marrow, and the phenotype of the OT-II cells was analyzed at six weeks of age. Compilation of the proportion of CD25⁺FoxP3⁺ T_{reg} among the OT-II Tg cells in the thymus or the lymph nodes.

3) In Figure 4, two major issues: a) Why no negative selection? Need to see flow data for CD4/CD8 stain in thymus and OT-II clonotype. B) Using % of CD4's is not enough, need to show absolute numbers of Treg's. Previous work, such as in Ref. #22 has shown that when deletion and Treg induction are mixed together absolute numbers of Treg's are unchanged in OT-II x RipMova mice.

As per the reviewer's request, we now include the flow cytometry profiles for CD4/CD8 as well as the absolute numbers for the OT-II Tg T_{reg} numbers. It is possible that we see a decrease in model antigen-specific cells in our tetramer analysis, but not with the OT-II model due to differences in TCR affinity for the model antigen coupled with relatively low INS2 transgene expression in the thymus. In fact, the OT-II TCR may be of lower affinity than many of the polyclonal cells¹². In addition, our results are consistent with other studies that reported that high numbers of cells expressing self-antigen such as ubiquitous antigen was required to drive clonal deletion, whereas lower thymic antigen presentation such as the one of tissue restricted antigen predominantly leads to non-deletional tolerance mechanisms and regulatory T cell (T_{reg}) generation¹³.

The OT-II cells were identified using the Thy1.1 congenic marker.

4) For CRP Tg mice: Which TEC cells are doing the selection? How can you really tell?

With the data initially presented, it was difficult to definitively determine which TEC were driving selection in CRP Tg mice. To clarify this point, we sorted mTEC^{lo} and mTEC^{hi} cells from the CRP Tg mice and co-cultured them with WT and OT-I thymocytes. By doing this, we were able to determine that both mTEC^{lo} and mTEC^{hi} cells from CRP Tg mice are able to induce negative

selection of antigen specific thymocytes. Interestingly, and in line with our hypothesis, we observed that differentiation of antigen-specific TCR β^+ CD4 $^+$ CD8 $^-$ DN cells is higher when OT-I thymocytes are cultured with mTEC $^{\text{lo}}$ than with mTEC $^{\text{hi}}$ (4.3 vs 2.8 fold increase). These results are included in the revised manuscript (Supplementary Fig. 3e and f).

5) In polyclonal mice can thymi be pooled from multiple mice to look at tetramer positive cells??

We did attempt to analyze tetramer+ cells in the thymus of individual mice, but the number of cells was too low that we were not confident in the data. While it is possible to pool multiple mice for these experiments, the amount of reagents needed for these experiments is quite significant considering that replicates also need to be performed. We hope the reviewer agrees that we have provided significant additional data in support our original conclusions and further identify the role of Aire in driving T $_{\text{reg}}$ development in INS2 Tg mice.

Minor comment:

Figure 5: What are the PE positive cells that are APC negative in panel f?

Both PE- and APC-conjugated tetramers were tested/titrated beforehand by staining SMARTA Tg T cells diluted in wild-type polyclonal splenocytes to ensure their quality. The population of tetPE+/tetAPC- cells that was evident in the original panel is an artifact of the PE enrichment that was performed. As such, only the double positive cells are considered true antigen-specific cells.

More specifically, a few of our experiments included a PeCy7-labeled antibody in the mix, and it is possible that the PeCy7 antibody binds to the PE beads during the PE enrichment leading to the appearance of a PE positive population due to some fluorochrome uncoupling. This PE $^{\text{hi}}$ single positive population only occurs in the presence of the PeCy7 antibody. We have now replaced these plots with representative profiles from an I-A $^{\text{b}}$ -gp66 tetramer staining experiment in which no PeCy7-conjugated antibody was included. The inclusion/exclusion of the PeCy7-conjugated antibody did not influence the results of this experiment.

Figure 5: Cell numbers in panel H are very low (as expected), how many mice were used? Is this really statistically significant? What about absolute numbers of cells (this is very important)? Finally, are these iTreg's vs thymic Treg's???

Mice were not pooled together to perform the tetramer analysis and a total of 6-10 CRP and 7-8 INS2 Tg mice (and 6-10 and 7-8 LMC were used, respectively).

Absolute cell numbers were different between littermate controls from the different TRA Tg lines; thus, to be able to compare the CRP and INS2 Tg mice, we normalized the absolute number of tetramer+ cells to that of their respective littermate controls. To be clear, the absolute numbers (not the proportions) of cells were used for these calculations.

We have now also included the relative number of gp66-specific T $_{\text{reg}}$ to Fig. 5 in addition to the proportion of gp66-specific T $_{\text{reg}}$ cells. As detailed in the response to Reviewer 3, comment 1, we suspect that these T $_{\text{reg}}$ are largely thymic-derived as adoptive transfer of OT-II or SMARTA T cells do not give rise to an increased proportion of T $_{\text{reg}}$ in INS2 Tg mice after LCMV-OVA infection. In addition, there is no increase in T $_{\text{reg}}$ number in Aire $^{-/-}$ INS2 mice in which the transgene expression is lost in the thymus but still present in the periphery suggesting, again, that they are thymically derived T $_{\text{reg}}$.

- 1 Zhang, Z., Legoux, F. P., Vaughan, S. W. & Moon, J. J. Opposing peripheral fates of tissue-restricted self antigen-specific conventional and regulatory CD4(+) T cells. *Eur J Immunol*, doi:10.1002/eji.201948180 (2019).
- 2 Xiao, Z., Mescher, M. F. & Jameson, S. C. Detuning CD8 T cells: down-regulation of CD8 expression, tetramer binding, and response during CTL activation. *J Exp Med* **204**, 2667-2677, doi:10.1084/jem.20062376 (2007).
- 3 Wooldridge, L. *et al.* Anti-coreceptor antibodies profoundly affect staining with peptide-MHC class I and class II tetramers. *Eur J Immunol* **36**, 1847-1855, doi:10.1002/eji.200635886 (2006).
- 4 Wang, R., Natarajan, K. & Margulies, D. H. Structural basis of the CD8 alpha beta/MHC class I interaction: focused recognition orients CD8 beta to a T cell proximal position. *J Immunol* **183**, 2554-2564, doi:10.4049/jimmunol.0901276 (2009).
- 5 Sun, J. & Kavathas, P. B. Comparison of the roles of CD8 alpha alpha and CD8 alpha beta in interaction with MHC class I. *J Immunol* **159**, 6077-6082 (1997).
- 6 Angelov, G. S., Guillaume, P. & Luescher, I. F. CD8beta knockout mice mount normal anti-viral CD8+ T cell responses--but why? *Int Immunol* **21**, 123-135, doi:10.1093/intimm/dxn130 (2009).
- 7 Weist, B. M., Kurd, N., Boussier, J., Chan, S. W. & Robey, E. A. Thymic regulatory T cell niche size is dictated by limiting IL-2 from antigen-bearing dendritic cells and feedback competition. *Nat Immunol* **16**, 635-641, doi:10.1038/ni.3171 (2015).
- 8 Bautista, J. L. *et al.* Intraclonal competition limits the fate determination of regulatory T cells in the thymus. *Nat Immunol* **10**, 610-617, doi:10.1038/ni.1739 (2009).
- 9 Leung, M. W., Shen, S. & Lafaille, J. J. TCR-dependent differentiation of thymic Foxp3+ cells is limited to small clonal sizes. *J Exp Med* **206**, 2121-2130, doi:10.1084/jem.20091033 (2009).
- 10 Kurd, N. & Robey, E. A. T-cell selection in the thymus: a spatial and temporal perspective. *Immunol Rev* **271**, 114-126, doi:10.1111/imr.12398 (2016).
- 11 Nadia, S. K., Ashley, H., Jaewon, Y., Brian, M. W. & Ellen, A. R. Factors that influence the thymic selection of CD8 $\alpha\alpha$ intraepithelial lymphocytes. *bioRxiv*, doi:<https://doi.org/10.1101/761601> (2019).
- 12 Mandl, J. N., Monteiro, J. P., Vriskoop, N. & Germain, R. N. T cell-positive selection uses self-ligand binding strength to optimize repertoire recognition of foreign antigens. *Immunity* **38**, 263-274, doi:10.1016/j.immuni.2012.09.011 (2013).
- 13 Malhotra, D. *et al.* Tolerance is established in polyclonal CD4(+) T cells by distinct mechanisms, according to self-peptide expression patterns. *Nat Immunol* **17**, 187-195, doi:10.1038/ni.3327 (2016).

REVIEWERS' COMMENTS:

Reviewer #1 (Remarks to the Author):

The authors have done a good job in addressing my previous concerns, in particular by acknowledging that besides mTEC subset specific expression, the expression level in individual cells and the frequency of cells expressing a given antigen may (in parallel/together!) determine to the outcome of thymic selection. Having said this, I am still not convinced that the MS conclusively supports the claim that distinct subsets of mTECs 'generally' induce distinct modes of central tolerance. However, it adds a perhaps controversial piece of information to a longstanding question in the field and in my opinion deserves publication in Nat Comm.

Reviewer #2 (Remarks to the Author):

The authors have substantially revised their manuscript, and the inclusion of new and extended data improves it and addresses all of the comments I raised.

Reviewer #3 (Remarks to the Author):

This is a revised version of a MS from Melichar and colleagues that attempts to connect thymic expression of self-antigen with T cell selection fate. From what I can tell the authors are proposing that mTEChi (via their Ins2 BAC Tg model) cells promote Treg selection for CD4 T cells when compared to mTEClo cells (via their CRP BAC Tg model). In the case of the latter, T cells can be pushed into the CD8aa lineage. Here in the revised MS, the authors have included new single cell RNA-Seq data on TEC's and additional data related to my original concerns and they should be commended for the hard work and effort that they have put in here to deal with these concerns.

In the end, my conceptual concern for this MS remains. Again, there is a host of literature on TEC populations inducing deletion or Treg selection in the thymus, but a simple model that a single thymic APC cell population promotes expression of Treg's vs deletion is probably just that, too simple. Jenkins and colleagues in their Nature Immunology paper in 2016 examined GFP-specific T cells in the polyclonal repertoire and came to the conclusion that limited expression within the thymus could promote Treg induction vs deletion (Nat Immunol 2016 Feb;17(2):187-95). Likewise, as stated in my original review, Aire-driven BAC transgene expression in mTEC hi can drive CD4+ Treg induction as shown in Nat Immunol. 2007 Apr;8(4):351-8. And again as I stated in original review the affinity of the TCR for its cognate antigen will lead to different outcomes (see Savage et al or Hsieh et al. Immunity papers). Therefore, many of the results here are observing similar phenomenon that has been described before with little new on the mechanism(s) by which these selection events are controlled. Indeed, the data here do not conclusively prove the title of the paper related to all of these concerns I have outlined here.

Minor comment:

The addition of the new experiment of Single Cell RNA-Seq in Figure 1 provides little clarity to the expression of their Ins2 BAC Tg or the CRPlo BAC Tg. There is no data here to show that endogenous Insulin 2 is expressed in the exact same cells as the USA driven off the Ins2 BAC. The statement : "the pattern of expression recapitulates that of their endogenous genes counterparts" is simply not supported by the new single cell experiment. Indeed, if it were, the USA driven off the Ins2 BAC

should be detectable in the exact same cells that are Ins2 positive in the UMAP plots and at similar levels for example.

Reviewer #1 (Remarks to the Author):

The authors have done a good job in addressing my previous concerns, in particular by acknowledging that besides mTEC subset specific expression, the expression level in individual cells and the frequency of cells expressing a given antigen may (in parallel/together!) determine to the outcome of thymic selection. Having said this, I am still not convinced that the MS conclusively supports the claim that distinct subsets of mTECs 'generally' induce distinct modes of central tolerance. However, it adds a perhaps controversial piece of information to a longstanding question in the field and in my opinion deserves publication in Nat Comm.

We thank the reviewer for the thoughtful critiques that improved our manuscript during this process. We agree with the reviewer's final remarks, though we believe that additional experiments to further address these issues fall outside the scope of this manuscript. In follow-up studies, we certainly seek to determine the extent to which endogenous self-antigens that are differentially expressed among mTEC subsets support distinct modes of T cell tolerance; this is not straight-forward and will take considerable effort. Our revised manuscript tones down the conclusions reached based on our analysis of the novel transgenic models created for this study. Further, in the Discussion, we highlight the need, going forward, for experiments to assess to what extent these observations are generalizable to the mode of T cell tolerance induction across additional Aire-dependent and -independent TRAs.

Reviewer #2 (Remarks to the Author):

The authors have substantially revised their manuscript, and the inclusion of new and extended data improves it and addresses all of the comments I raised.

We thank the reviewer for the constructive critiques of our original manuscript that helped us to improve its content and clarity.

Reviewer #3 (Remarks to the Author):

This is a revised version of a MS from Melichar and colleagues that attempts to connect thymic expression of self-antigen with T cell selection fate. From what I can, tell the authors are proposing that mTEChi (via their Ins2 BAC Tg model) cells promote Treg selection for CD4 T cells when compared to mTEClo cells (via their CRP BAC Tg model). In the case of the latter, T cells can be pushed into the CD8aa lineage. Here in the revised MS, the authors have included new single cell RNA-Seq data on TEC's and additional data related to my original concerns and they should be commended for the hard work and effort that they have put in here to deal with these concerns.

Indeed, we made a considerable effort to address the thoughtful reviews of the referees that certainly helped to improve our manuscript.

In the end, my conceptual concern for this MS remains. Again, there is a host of literature on TEC populations inducing deletion or Treg selection in the thymus, but a simple model

that a single thymic APC cell population promotes expression of Treg's vs deletion is probably just that, too simple. Jenkins and colleagues in their Nature Immunology paper in 2016 examined GFP-specific T cells in the polyclonal repertoire and came to the conclusion that limited expression within the thymus could promote Treg induction vs deletion (Nat Immunol 2016 Feb;17(2):187-95). Likewise, as stated in my original review, Aire-driven BAC transgene expression in mTEC hi can drive CD4+ Treg induction as shown in Nat Immunol. 2007 Apr;8(4):351-8. And again as I stated in original review the affinity of the TCR for its cognate antigen will lead to different outcomes (see Savage et al or Hsieh et al. Immunity papers). Therefore, many of the results here are observing similar phenomenon that has been described before with little new on the mechanism(s) by which these selection events are controlled. Indeed, the data here do not conclusively prove the title of the paper related to all of these concerns I have outlined here.

We understand the reviewer's points and have further updated the text to address the remaining issues. First, we have revised the text to make it clear that our manuscript is focused on T cell-extrinsic variables that impact autoreactive thymocyte fate and that mTEC subset is likely one of several variables that include TCR affinity for self-antigen and the number of thymic APC expressing a given self-peptide that influence the mechanism of T cell tolerance. Further, with suggestions from the editor, we have also modified the title and text to avoid over-stating the conclusions of our study.

In Aschenbrenner et al, Nat Immunol 2007, several points that are particularly relevant to our current study were made. In particular, in the AIRE-promoter driven model antigen transgenic models that the reviewer alludes to, (Aire and) model 'self antigens' are expressed in the sorted mTEC_{lo} and mTEC_{hi} subsets. Although the authors find that model antigen expressing mTEC_{lo} only minimally stimulated T cell hybridoma activation, there was little comparison of tolerance induced by mTEC_{lo} versus mTEC_{hi} subsets in this article. In that paper's forward-looking Discussion, the authors state that "... functional heterogeneity in the mTEC compartment (for example, varying amounts of costimulatory molecules) may influence the outcome of deletion versus T_{reg} cell fate specification." There is no dispute that this group developed useful models and made an important contribution to the field particularly as it relates to MHC II-restricted T cells. We more clearly highlight this work in the context of our findings in the discussion.

We believe that our study offers substantial insight complementary to that of by Aschenbrenner et al and the other important references that the reviewer identifies. This includes bringing new tools to the field (BAC Tg models with mTEC_{lo}-biased and mTEC_{hi}-restricted model antigen expression), and identifying potential differences between tolerance induction by mTEC_{lo} and mTEC_{hi} subsets in terms of negative selection/Treg development, while also providing deeper understanding of the fate of self-reactive MHC I-restricted thymocytes in these models.

Minor comment:

The addition of the new experiment of Single Cell RNA-Seq in Figure 1 provides little clarity to the expression of their Ins2 BAC Tg or the CRP_{lo} BAC Tg. There is no data here to show that endogenous Insulin 2 is expressed in the exact same cells as the USA driven off the Ins2 BAC. The statement : "the pattern of expression recapitulates that of their endogenous genes counterparts" is simply not supported by the new single cell experiment. Indeed, if it were, the USA driven off the Ins2 BAC should be detectable in the exact same cells that are Ins2 positive in the UMAP plots and at similar levels for example.

This is a relevant point by the reviewer. We have edited the text to clarify that although CRP and INS2 transgene expression is detected in fewer cells than their endogenous counterparts, their broad pattern of expression among mTEC subsets is similar to the *Crp* and *Ins2* genes, respectively.